# Research Progress of Horizontal Cavity Surface-Emitting Laser

**DOI:** 10.3390/s23115021

**Published:** 2023-05-24

**Authors:** Jishun Liu, Yue Song, Yongyi Chen, Li Qin, Lei Liang, Shen Niu, Ye Wang, Peng Jia, Cheng Qiu, Yuxin Lei, Yubing Wang, Yongqiang Ning, Lijun Wang

**Affiliations:** 1State Key Laboratory of Luminescence and Applications, Changchun Institute of Optics, Fine Mechanics and Physics, Chinese Academy of Sciences, Changchun 130033, China; liujishun21@mails.ucas.ac.cn (J.L.); qinl@ciomp.ac.cn (L.Q.); liangl@ciomp.ac.cn (L.L.); niushen20@mails.ucas.ac.cn (S.N.); wangye@ciomp.ac.cn (Y.W.); jiapeng@ciomp.ac.cn (P.J.); qiucheng@ciomp.ac.cn (C.Q.);; 2Daheng College, University of Chinese Academy of Sciences, Beijing 100049, China; 3Jlight Semiconductor Technology Co., Ltd., Changchun 130033, China; 4College of Opto-Electronic Engineering, Changchun University of Science and Technology, Changchun 130022, China; 5Peng Cheng Laboratory, No. 2 Xingke 1st Street, Nanshan, Shenzhen 518000, China; 6Academician Team Innovation Center of Hainan Province, Key Laboratory of Laser Technology and Optoelectronic Functional Materials of Hainan Province, School of Physics and Electronic Engineering, Hainan Normal University, Haikou 570206, China

**Keywords:** surface emission laser, SE-DFB laser, photonic crystal, second order diffraction, high power semiconductor laser

## Abstract

The horizontal cavity surface emitting laser (HCSEL) boasts excellent properties, including high power, high beam quality, and ease of packaging and integration. It fundamentally resolves the problem of the large divergence angle in traditional edge-emitting semiconductor lasers, making it a feasible scheme for realizing high-power, small-divergence-angle, and high-beam-quality semiconductor lasers. Here, we introduce the technical scheme and review the development status of HCSELs. Firstly, we thoroughly analyze the structure, working principles, and performance characteristics of HCSELs according to different structures, such as the structural characteristics and key technologies. Additionally, we describe their optical properties. Finally, we analyze and discuss potential development prospects and challenges for HCSELs.

## 1. Introduction

Semiconductor lasers have garnered significant interest among researchers due to their advantages, such as small size, long lifespan, high conversion efficiency, fast modulation rate, wide wavelength range, and ease of integration [1,2]. Over the course of their development, semiconductor lasers have found extensive use in various fields, including industrial production, military defense, and medical treatment [3,4,5]. Furthermore, they have emerged as crucial light sources in optical communication, optical pump lasers, and optical information storage [6]. With the progression of society and the expansion of semiconductor laser application in laser radar, laser processing, and high-speed laser communication, individuals require laser light sources with lower divergence angles, higher power, and higher slope efficiency [7,8]. Nonetheless, conventional semiconductor lasers have drawbacks such as susceptibility to cavity surface damage [9], significant divergence, and inadequate monochromaticity [10,11]. These conventional semiconductor lasers require complex systems for beam shaping, collimation, and coupling to attain high beam quality, but their high cost renders them inadequate for affordable use.

Commercial vertical-cavity surface-emitting semiconductor lasers (VCSELs) have superior performance with excellent beam shape, no cavity surface catastrophe damage, and easy two-dimensional integration [12]. However, the thin active region of VCSEL results in a lower single-way gain [13], which limits its output power severely. Even with lateral multimode, the competition from higher-order modes can degrade beam quality substantially [14,15].

Horizontal-cavity surface-emitting semiconductor lasers (HCSELs) provide the advantages of high power and high coupling efficiency [16]. The structure of HCSEL introduces a diffraction or reflection structure that improves the beam quality so that the light is emitted in a direction perpendicular to the epitaxial surface of the crystal. The realization of the longitudinal mode characteristics of the surface-emitting laser is limited by the diffraction structure, which in turn has the advantages of small temperature drift and narrow spectrum [17]. Moreover, the power-bearing capacity of crystal epitaxial surface is higher than a cleaved or etched cavity surface, making it possible to withstand higher single-mode output power [18]. HCSEL also offers benefits such as high surface damage threshold, simple manufacturing, and easier 2D array integration [19,20]. Therefore, this paper briefly describes the working principle, device structure, research progress, and development trends of three different HCSELs.

## 2. Structure and Principle of the HCSEL

There are three main approaches for achieving high-power horizontal cavity surface-emitting: mirror emission, grating diffraction, and utilizing the periodic dielectric constants of photonic crystals to accomplish surface light emission.

### 2.1. Grating Diffraction

A distributed feedback (DFB) grating structure is introduced to diffract the laser light from the horizontal to the vertical direction, achieving surface lasing [21]. Figure 1 shows the basic structure diagram of a second-order grating surface emission DFB (SE-DFB) laser as an example. The laser device has a second-order grating etched on its P-limited layer. Laser light is emitted from surfaces by diffraction through a waveguide layer with a second-order grating. Additionally, Bragg grating is utilized for DFB of the light wave, resulting in a single longitudinal mode.

The laser’s surface emission and stable light oscillation in the resonant cavity are achieved by satisfying the diffraction and resonance conditions, respectively.

The diffraction condition of the grating can be expressed as:(1)$$\sin\varphi =\sin\theta +\frac{m\lambda }{n_{\mathrm{eff}}*\Lambda },$$

These parameters include θ as the incident angle, φ as the scattering angle, m as the diffraction order, which is indicated by an integer, λ as the wavelength of the incident light, n_eff_ as the effective refractive index, and Λ as the grating period.

The resonance condition for the grating can be expressed as follows:(2)$$\Lambda =\frac{{m*\lambda }_{B}}{2{*n}_{\mathrm{eff}}},$$

The grating period is denoted by Λ, the Bragg wavelength by λ_B_, the effective refractive index of the grating by n_eff_, and the diffraction order by m (when m = 2, the grating operates at its resonance condition). By combining Formulas (1) and (2), we obtain a second-order grating that diffracts the first-order light perpendicular to the incident light and the second-order light in the opposite direction. Figure 2 depicts a schematic diagram of these diffraction characteristics.

The first-order diffracted light is responsible for output coupling and influences the surface emission coupling coefficient. The second-order diffracted light is utilized for optical feedback and mode selection, which in turn impacts the feedback coupling coefficient. The laser’s light output characteristics are influenced by the combination of these two coupling coefficients. Laser stability is achieved through an increase in the surface emission coupling coefficient, resulting in a larger threshold gain difference between the main mode and the lowest order mode, eliminating the degeneracy of modes, and selecting a single longitudinal mode to stabilize the laser wavelength.

### 2.2. Photonic Crystal Diffraction Structure

Photonic crystal structures have applications in both horizontal cavity lasers and vertical cavity lasers [22]. Among them, the HCSELs utilize the photonic crystal diffraction of the photonic crystal band-edge mode and rely on the band-edge resonance effect of the two-dimensional photonic crystals to produce stimulated emission amplification and surface emission [23]. These types of lasers are commonly referred to as photonic crystal surface-emitting lasers (PCSEL). The photonic crystal (PC) can be viewed as a two-dimensional grating structure, similar to the lasing principle of SE-DFB lasers.

Figure 3 depicts the energy band diagram of a two-dimensional square lattice air hole photonic crystal slab. The gray region in the diagram represents the light cone. At high symmetry points x, M, and Г, the slopes are almost zero. This can be observed from the group velocity formula:(3)$$v_{g}=\frac{d\omega }{\mathrm{dk}},$$

The analysis demonstrates that a low group velocity results in longer interaction time and more effective coupling between the radiation mode and the material system, leading to enhanced optical processes, such as stimulated emission, nonlinear optical processes, and light absorption. Consequently, these modes are optically localized and form standing wave oscillations at the high symmetry point of the photonic crystal energy band. A mode inside the light cone may be a leaky mode from the photonic crystal resonator, with surface-emitting lasing properties. Conversely, the modes under the light cone are confined within the photonic crystal resonator [24].
Figure 3Energy band diagram of a two-dimensional square lattice photonic crystal slab structure [25]. © The American Association for the Advancement of Science. Copyright 2001 Science.
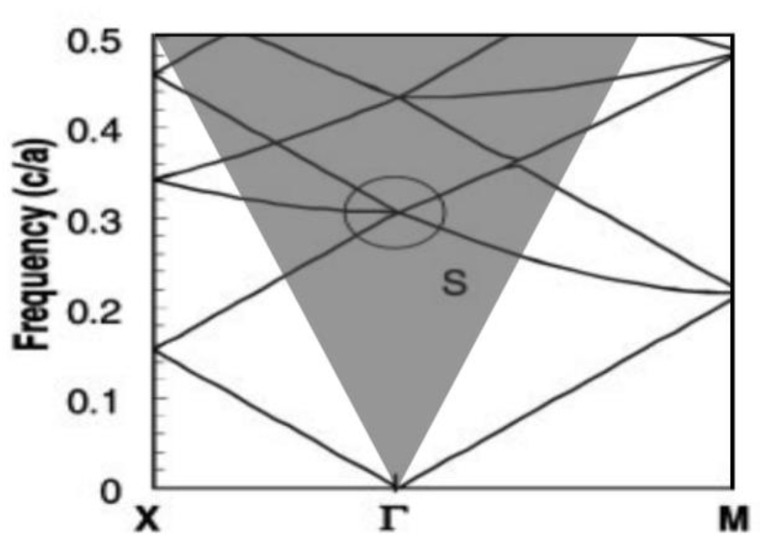



As shown in Figure 4, the laser resonator is composed of an array of photonic crystals arranged in a two-dimensional square lattice. The spacing of the photonic crystals in the array matches the lasing wavelength in both the x and y directions, satisfying the second-order Bragg diffraction condition, which diffracts light propagating in the +y direction to the -y direction. Furthermore, since it meets the first-order Bragg diffraction condition, light waves are diffracted into the ±x direction as well. Ultimately, the light waves propagating in these four directions interact, leading to the formation of a two-dimensional standing wave and a large-area photonic crystal cavity. If the plane satisfies the first-order Bragg diffraction condition, the emission will diffract vertically to create surface emissions. This phenomenon can be used to produce a photonic crystal surface-emitting laser with a large area of light output and a small divergence angle [26].

### 2.3. Mirror-Type

Steering mirror-type horizontal cavity surface-emitting semiconductor lasers [27,28] incorporate a steering mirror onto the photonic chip and etch a grating structure above the active layer. The DFB structure is used to realize the mode selection and optical feedback of the laser, and the optical path is changed by integrating a 45° reflector horizontally to realize surface emission (Figure 5). The mode selection of the structure to the light wave is similar to that of the surface emitting DFB (SE-DFB) laser, which is determined by the resonance condition of the second order grating. At the same time, the surface emission of the laser relies on a 45° reflector, which means that the smoothness and quality of the reflector determine the output power. Thus, the etching process of this structure on the steering mirror is critical.

## 3. Research Progress of HCSEL

The evolution of HCSELs dates back to the late 19th and early 20th centuries and aims to overcome the high divergence angles present in edge-emitting lasers. Diffractive structures or etched mirrors on the chip are commonly employed to facilitate surface emission in HCSELs. Table 1 presents the research advancements of HCSELs over the last decade.

### 3.1. SE-DFB Laser

#### 3.1.1. Linear Second-Order Grating

In 2005, the University of Wisconsin developed a surface-emitting laser with a second-order grating structure that is free from aluminum [29]. The structure employs preferential etching and secondary epitaxy techniques for the purpose of controlling the transverse mode. The output light of this laser has a wavelength of 980 nm, and the grating structure is partitioned into a 700 μm second-order DFB grating and a 600 μm distributed Bragg reflector (DBR) structure on both sides. The structure presents a transverse effective refractive index difference, with a 20-unit array resulting in continuous and pulsed output powers of 1.6 W and 10 W, consecutively.

In 2007, Sushil Kumar et al. [41] from the Massachusetts Institute of Technology reported a surface emission DFB terahertz (THz) quantum cascade single-mode laser, and obtained robust single-mode operation and single-lobe beam mode in this structure. They used a second-order DFB grating to achieve surface coupling out of the laser beam. Additionally, they combined a metal–metal DFB structure with a resonant phonon THz quantum cascade laser (QCL) active region. With this structure, not only can the low waveguide loss advantage of the metal–metal waveguide be maintained but its output power can also be increased and the beam quality improved. The frequency range of the quantum cascade single-mode laser is 0.35 THz~2.9 THz. The continuous wave power of the laser reaches 6 mW at 5 K. Additionally, the maximum pulsed operating temperature can reach 149 K.

In 2014, the University of Wisconsin carried out the design of a grating-coupled surface-emitting QCL that operates in the 4.6 μm band [42]. This laser has the remarkable capability of suppressing the antisymmetric mode, which enables symmetric mode lasing at a low threshold current and high slope efficiency. The device relies on the feedback from the light wave mode and the second-order distribution to initiate the resonant coupling function of the antisymmetric surface plasmon mode of the metal–semiconductor grating, thereby suppressing the antisymmetric mode and enhancing the symmetric mode significantly. The device features distributed Bragg reflector (DBR) gratings at both ends, which contain the optical field and carrier distribution. As a result, the uncontrolled reflection of the cleavage facet and cavity mirror degradation can be avoided when operating at high power. Consequently, the device has the potential to generate a stable and coherent output of light, with the output power capable of reaching a level of watts when in continuous wave operation. Specifically, the 7.03 mm long grating-coupled surface-emitting QCL exhibits a remarkably low threshold current of 0.45 A and an impressive slope efficiency of 3.4 W/A.

In 2016, researchers from Taiwan [32] presented the InAs/InGaAs QD SE-DFB laser for the first time. They utilized indium–tin–oxide (ITO) as the p-type cladding layer. The benefits of using ITO include the avoidance of the annealing effect of high-temperature epitaxial growth on the quantum dots due to its lower growth temperature compared to the deposited quantum dots. Furthermore, its transparency yields low-loss vertical light output while still providing excellent ohmic contact. In addition, the relatively low refractive index of ITO provides good light confinement. The study indicates that ITO can potentially replace traditional semiconductor cladding, simplifying the semiconductor laser process technology. This research developed a QD SE-DFB laser in the 1.3 μm band with a threshold current density of 210 A/cm^2^, a characteristic temperature T_0_ of 94 K, and a temperature-dependent wavelength shift of only 0.1 nm/K. The wavelength stability is six times greater than conventional Fabry–Perot (FP) lasers, indicating good temperature properties.

In 2017, the Institute of Semiconductors at the Chinese Academy of Sciences [33] developed a surface-emitting QCL array with a coupled ridge-waveguide structure operating at a wavelength of 7 μm. The array consists of 15 parallel elements for increased peak power. The coupled ridge waveguide structure produces a narrow divergence angle of 2.9° in the ridge width direction and 0.36° in the cavity length direction, resulting in a single-lobe far-field pattern. The use of a buried grating structure ensures stable single longitudinal mode emission and high extraction efficiency. At 25 °C, the surface-emitting DFB QCL array achieved a peak output power exceeding 2 W, while at a temperature of 15 °C, and the maximum output power was 2.29 W and the slope efficiency was approximately 500 mW/A.

In 2017, the Institute of Semiconductors, Chinese Academy of Sciences [34] designed a buried grating-coupled substrate emitting DFB QCL. This design uses a semiconductor buried second-order grating structure to enable in-plane feedback and the vertical coupling of semiconductor grating. This structure can eliminate the uncontrollable reflection of the cleavage surface without the integrated DBR grating. A low-loss symmetrical mode continuous laser is produced that can emit up to 248 mW of power at the surface while operating continuously in the 4.97 µm band (operating temperature is 20 °C). Under the action of injected current, a good single-mode characteristic is obtained, and the far-field divergence angle is approximately 0.14° × 16°.

In 2018, a research team from Changchun University of Science and Technology [43] explored the utilization of an asymmetric waveguide structure to enhance the performance of a second-order diffraction SE-DFB semiconductor laser. The study investigated a structure composed of varying thicknesses of the P and N waveguide layers, while simulating the performance of the structure. Through simulation, the researchers found that the asymmetric waveguide structure greatly influences the distribution of light by increasing the confined light field and promoting photon interaction with the second-order grating. As a result, this approach successfully reduced the threshold current of the improved horizontal resonator surface emission SE-DFB semiconductor laser. The laser has an output power of 100 mW, a slope efficiency of 1.04 W/A, with a chip size of 500 μm × 1000 μm, and an exit hole area of 80 μm × 4 μm.

In 2018,the Institute of Semiconductors, Chinese Academy of Sciences developed a second-order grating surface-emitting THz QCL based on metal–metal waveguides [44]. With a maximum peak power of 12.2 mW in pulsed mode, the laser achieves a maximum slope efficiency of 60.4 mW/A at 5 K and can operate at a maximum temperature of 105 K. The emissions achieve a stable single-mode at approximately 89 μm in wavelength, with a side-mode suppression ratio of about 25 dB under all operating conditions. Furthermore, a single-lobe far-field radiation pattern with a small divergence angle of 4° in the cavity length direction can be achieved by inserting a central phase shift in the device.

In 2020, the Institute of Solid-State Electronics of the Vienna University of Technology [45] reported the development of an interband cascade laser (ICL). The laser has the capacity to emit light from surfaces in continuous-wave mode at 38 °C. Its unique ring cavity structure (Figure 6) with a downward mounting on the exterior surface, efficiently extracts heat from the device. A second-order DFB grating is also employed for single-mode emission at 3.8 μm wavelength in a vertical orientation. A ring with an 800 μm diameter generates light output power exceeding 6 mW at 20 °C. The threshold current density reaches 0.6 kA/cm^2^.

In 2021, researchers at the Changchun University of Science and Technology conducted research [39] on a new kind of second-order grating SE-DFB semiconductor laser. By emitting from the p-surface, the laser achieved a narrow linewidth and reduced far-field divergence angle, which was presented in Figure 7. Unlike the conventional n-substrate emitting horizontal cavity SE-DFB semiconductor lasers, the p-surface emitting laser was fabricated without secondary epitaxy and double-side alignment processes, making the fabrication process more straightforward. The laser operated at a wavelength of 976.2 nm and had an output power of 86mW. It boasted a 0.94 nm linewidth and 2.6° × 6.1° far-field divergence angle, accompanied by a threshold current of 450 mA.

In 2022, Tian kun et al. [46] of Changchun University of Science and Technology found that neglecting the treatment of non-output radiated light leads to serious power loss, resulting in issues like low differential quantum efficiency and mode instability through simulation. Traditional metal electrodes at the non-output radiated light source provide some reflectivity. However, the smoothness of electrodes affects the reflection and scattering of light, demanding high levels of thinning accuracy on the wafer. The regular wafer grinder cannot achieve the required thinning accuracy. Therefore, to address this challenge, they proposed the fabrication of distributed Bragg reflector (DBR) multilayer reflectors on the P-side second-order grating to enhance the directivity of the device (Figure 8) and simulated the structure. Specifically, they etched the surface grating into the P-side cladding layer and coated it with a thick Si_3_N_4_ film. A SiO_2_/Si_3_N_4_ multilayer reflector was formed on the thick film to reflect the upward-diffracted light. This approach offers several benefits, such as avoiding growth interruption and re-growth of the epitaxial structure, simplifying the device manufacturing process, and reducing preparation costs. In addition, the high-reflectivity mirrors based on dielectric materials offer flexibility in adjusting the device structure because they do not depend on epitaxial materials. Moreover, the radiation directionality of the surface of the structure is strong, which enhances the device’s performance and causes the high directionality of the surface radiation to reach over 98%.

#### 3.1.2. Curved Second Order Grating

In 2010, the American Alfalight Company [47] developed a SE-DFB laser with a curved second-order grating to enhance the output power significantly. The curved grating creates an unstable resonator cavity design, enabling lateral mode control, resulting in higher brightness when compared to edge-emitting lasers. This technology enables a single laser diode to supplant many of the more complex solid-state lasers. The continuous output power from a single transmitter was 73 W, with peak pulsed power exceeding 300 W.

In 2012, the Alfalight Company [30] employed InP-based materials to produce curved second-order gratings intended to enhance the output power of SE-DFB lasers. The grating is engraved on the p-cladding layer, with a pumping area at its center, a non-pumping area and an absorption area on its edges to suppress Fresnel reflection. The curved grating’s design constitutes a structure akin to a “traditional unstable cavity.” This arrangement allows the gain medium to provide energy to its maximum extent, resulting in a transverse mode with improved beam quality. The device has dimensions of 2.0 × 0.075 mm, making it a long-wavelength SE-DFB laser that can produce an operating output power higher than 1 W at room temperature (20 °C) with peak power conversion efficiency of 13%.

#### 3.1.3. High-Order Grating SE-DFB Laser

In 2018, Lehigh University [35] developed a THz QCL laser with a metal waveguide by replacing the second-order grating with a hybrid grating. The hybrid grating combines second-order and fourth-order Bragg gratings to excite symmetric modes with higher radiative efficiency. The second-order grating has a stronger DFB and helps establish resonant optical modes that have a similar phase relationship to the grating. The fourth-order grating enhances the outcoupling of the anti-symmetric mode and reduces the outcoupling of the symmetric mode. Adjusting the design parameters results in a DFB cavity with higher radiation efficiency compared to the excitation mode of the second-order DFB cavity. The researchers detected a peak power output of 170 mW with a slope efficiency of 993 mW/A for a 3.4 THz QCL operating at 62 K, using robust single-mode single-lobe emission. This study shows that the hybrid grating scheme is easier to implement than conventional DFB schemes and can increase the power output of surface-emitting DFB lasers at any wavelength.

In 2019, Huazhong University of Science and Technology [48] proposed a high-order surface grating Index-Coupled InGaAsP/InP SE-DFB laser at 1550 nm by combining edge-emitting lasers with second-order gratings. The device consists of a high-order surface grating etched on an edge-emitting first-order DFB laser to enable coupling of light from the laser cavity to the surface. Simulation results reveal a surface emission power of 4.2 mW at an injection current of 30mA, with the coupling efficiency of the fiber close to 50% and SMSR reaching 48 dB at this current. Compared to a second-order SE-DFB, the coupling efficiency of the six-order surface grating is only reduced by 5%, while the higher-order grating can be more deeply etched to attain higher output power. Additionally, the device inherits excellent performance and mature manufacturing processes from commercial edge-emitting DFB lasers and offers a means of achieving high-power surface-emitting lasers.

#### 3.1.4. Surface-Emitting DFB Lasers from New Semiconductor Materials

a. Perovskite SE-DFB laser

Perovskite DFB surface-emitting lasers have drawn increasing attention due to their solution processability, tunable bandgap, single-mode operation, and low threshold. These low-threshold characteristics have the potential to enable high-power surface-emitting lasers. However, fabricating high-quality DFB cavities on perovskite films has proven challenging, and overcoming this obstacle will be crucial for the development of low-threshold semiconductor lasers.

In 2018, the University of Texas at Dallas [49] reported the successful fabrication of SE-DFB methylammonium lead iodide (MAPbI_3_) lasers on silicon substrates. The perovskite film was patterned using thermal nanoimprint lithography (NIL) to achieve predefined cavity geometries, size control, repeatability, and high-Q-factor cavities with large mode gain overlap (Figure 9). This technique also improved the material’s emission characteristics. The perovskite laser demonstrated continuous laser output at room temperature and an ultra-low pump power density of 13 W/cm^2^. This research represents a significant step forward for the development of electrically pumped lasers in thin-film and organic materials and for the integration of perovskite lasers into photonic circuits.

In 2022, a facile method for fabricating DFB cavities was proposed at North Carolina State University [19]. The method involves adding polyvinylpyrrolidone to the perovskite precursor solution, which allows for the production of a stable and highly processable precursor film after spin coating. Lastly, a simple nanoimprint was achieved on the perovskite film using the preparation schematic presented in Figure 10. Furthermore, cavity design was obtained by analyzing the optical mode of the system, and by adjusting the solution concentration, the effective refractive index of the waveguide mode could be controlled. As a result, a perovskite DFB laser with a low threshold of 20 μJ/cm^2^ was obtained. This simplified fabrication process of a perovskite DFB laser cavity provides guidance for future studies on perovskite DFB lasers in the field.

b. Organic Semiconductor SE-DFB laser

In 2017, Kyushu University in Japan [50] successfully developed a surface-emitting organic DFB laser with low threshold. The study discovered that the organic lasing materials possess high photoluminescence quantum yield (PLQY) and optical gain with peak lasing emission and no triplet absorption (TA) bands’ spectral overlap, which makes them perfect for suppressing triplet losses. The research proves that, when excited with an 80 MHz quasi-continuous wave and 30 ms long pulse light, the surface-emitting organic DFB laser has no triplet absorption loss at the lasing wavelength, resulting in a low lasing threshold by utilizing a hybrid-order DFB grating. Accordingly, this research takes a significant step towards the development of genuine continuous wave organic laser technology, creating possibilities for realizing high-power surface-emitting organic semiconductor lasers on a large scale in the future.

In 2019, the University of Alicante in Spain [51] proposed an all-solution of high-performance to process organic DFB lasers. As shown in Figure 11, the solution involves a water-treated photo-resist layer, with a surface-relief grating placed on top of the active layer. The resonator design enables fine-tuning the device’s emission performance. Further, it was discovered that controlling the residual resist thickness and grating depth could enable the optimization of lasing threshold and efficiency simultaneously. The most exceptional efficiency and lowest threshold of the laser can be achieved by eliminating the residual layer, with the optimum depth of the grating found to be within the range of 100–130 nm.

In 2020, the University of Alicante in Spain [52] reported the development of lasers emitting within the spectral region ranging from 375 to 475 nm. The active material and resonator, depicted in Figure 12, were fabricated using solution-processable polymer films with a one-dimensional relief grating configuration. To create ten different active layers, the authors dispersed various organic compounds in polystyrene, including carbon-bridged oligo (p-styryl) derivatives and two fluorocarbon compounds. They successfully demonstrated the feasibility of fabricating a resonant cavity through holographic lithography, using a dichromate gelatin photo-resist on the active film.

c. Quasicrystal SE-DFB laser

Quasicrystal DFB lasers do not rely on mirror cavities to amplify and extract radiation. Upon implementation on the surface of a semiconductor laser, quasicrystal patterns can facilitate the tuning of radiative feedback and the extraction of high radiative and quality factor optical modes without defined symmetric or antisymmetric properties.

In 2020, Italian scholars [16] created a quasi-crystal THz QCL. The resonator incorporates a surface grating designed by Octonacci sequence (as shown in Figure 13) that greatly enhances the performance of surface-emitting THz lasers. Optimizing the interaction between the grating-scattered wave vector and photon propagation through the adjustment of the laser width and patterned slit size leads to efficient surface THz emission with a double-lobe beam profile. The researchers achieved a maximum peak optical power of 240 mW (190 mW) in a laser capable of both multimode and single-frequency modes, with a slope efficiency of 570 mW/A at 78 K and 700 mW/A at 20 K. The laser produced a high power output of THz radiation.

### 3.2. Photonic Crystal Diffraction Structure

#### 3.2.1. Single Lattice Structure

In 1999, Meier and Imada proposed surface emitting lasers using photonic crystal band edge modes [53,54]. Both research groups achieved band-edge surface emission based on the modulation of the energy band edge and lattice symmetry by the photonic crystal’s density of states. However, they have fundamental differences in their approach. Meier’s group utilized the band edge of the triangular-lattice photonic crystal but did not consider the Γ-point of the photonic crystal band edge. Consequently, there was a lack of two-dimensional coupling of light in the photonic crystal resonator, resulting in their device not exhibiting coherent two-dimensional (2D) oscillations. In contrast, Imada’s team improved the performance of their device by bonding the structure of the multi-quantum well active layer with a photonic crystal structure wafer and using the Г-point of the energy band edge in the triangular lattice photonic crystal. The light in the photonic crystal resonator undergoes one-dimensional and two-dimensional coupling simultaneously, resulting in true coherent oscillation. Additionally, the Г point characteristics were well-suited for vertical surface emission. Furthermore, Imada and his team observed consistent lasing at a wavelength of 1.3 μm across all tested areas, demonstrating successful realization of coherent oscillation over a large photonic crystal area.

In 2014, the Noda’s team and Hamamatsu’s team [31] developed a surface-emitting laser using a two-dimensional photonic crystal that was capable of generating high power output at a watt-level and good single-mode quality while under room temperature and continuous wave conditions. They utilized metal–organic chemical vapor deposition to form a photonic crystal that was capable of extending the coherent oscillation region by one thousand times (200 μm × 200 μm) through the two-dimensional band-edge resonance effect to enhance beam quality. In comparison to VCSELs, the focal spot was reduced by two orders of magnitude (M^2^ ≤ 1.1). The pore structure of the photonic crystal was vertically asymmetrical, resembling a triangular pyramid. The photonic crystal structure with a chip area of 200 μm × 200 μm was able to achieve a high-power continuous output of 1.5 W at room temperature.

In 2019, researchers from the Changchun Institute of Optics, Fine Mechanics and Physics [37] incorporated two-dimensional photonic crystals into their study and employed InAs quantum dots as their active material, as depicted in Figure 14. Compared to a VCSEL, this laser achieves surface emission through photonic crystal diffraction, circumventing the growth cost of DBR. Additionally, the enhanced output power is a result of the photonic crystal’s flat-band structure and an extra feedback mechanism. As a result of the study, a surface-emitting laser operating at a continuous wave with a wavelength of 1.3 μm at room temperature was developed. This device has an output of 13.3 mW of continuous power and 150 mW of pulsed power. Moreover, the device is able to operate in pulsed mode even at temperatures of up to 90 °C. This research offers a method to surpass the constraint of limited and low-power 1.3 μm surface-emitting lasers and expands its array of uses for higher-power consumption.

#### 3.2.2. Double-Lattice Structure

In 2018, the Noda team at Kyoto University in Japan [36] proposed a double-lattice structure, as shown in Figure 15. The dual-lattice PCESL was designed to enhance the vertical light output by minimizing resonances, except for the fundamental transverse mode, through the optimization of both the air filling factor and lattice points distance. The dual-lattice PCESL comprises cladding layers, sandwiching the PC and MQW layers, with p-type and n-type conductivity. The successful fabrication of the dual-lattice PCESL was performed using metal–organic vapor phase epitaxy (MOVPE) crystal growth and high-precision collimated electron beam lithography, in conjunction with two-step dry etching methods. The maximum output power of the dual-lattice PCESL was measured at 7 W with a continuous working current at 16 A, with a slope efficiency of approximately 0.48 W/A, while operating at a laser wavelength of 940 nm, which generated a divergence angle of less than 0.4°.

In the year of 2018, a research team [55] successfully produced a photonic crystal surface-emitting semiconductor laser with a diameter of 0.5 mm, a peak power of 10 W, and a far-field angle of less than 0.3°. By providing optical gain at the frequency of the band-edge cavity mode created in the photonic crystal, the device achieves in-plane wide-area coherent oscillation. By using appropriately designed PC air holes, the light was effectively diffracted both vertically and horizontally. The backward-diffracted air holes were designed to have an optical path difference of λ/2, which lead to in-plane destructive interference. The higher-order modes’ proximity to the edges leads to a faster increase in edge losses compared to the fundamental modes, thereby resulting in larger threshold gain margins. To increase surface emission, the researchers introduced a difference in the height of a pair of air holes, which created vertical asymmetry.

The following year, in 2019, the Noda team [56] published a theoretical study on a double-lattice photonic crystal resonator, as depicted in Figure 16. The double-lattice photonic crystal resonator operated on the principle of misaligning the two lattice point groups by λ/4 in both the horizontal and vertical directions. The equivalent double-lattice structure eliminates the direct 180° coupling between the fundamental Bloch waves while maintaining the indirect 90° coupling. Three double-lattice structures were compared, and their 180° diffraction coefficients κ_1D_ were calculated through simulation. The intensity values were I ~470 cm^−1^, II ~250 cm^−1^, and III ~100 cm^−1^ (Figure 17). The authors obtained the double-lattice structure III with the strongest 180° coupling suppression ability.

In 2020, the research team led by Noda [57] examined the thermal management system of large-area photonic crystals and designed its corresponding structure. The cooling system utilizes water and comprises a 1 mm thick copper base plate with a 300 μm thick submount attached to its upper surface. Water flows over the bottom surface of the copper base plate, while a fin structure on the same surface increases the area available for heat exchange. In addition, the study analyzed the distribution of power consumption by photonic crystal surface-emitting lasers. Figure 18 illustrates the calculated power consumption ratios of PCSELs with an 800 μm diameter at various slope efficiencies. It is apparent that the resonant cavity’s power consumption is relatively high when the slope efficiency is low. Thus, optimizing the radiation constant, tuning the number of quantum wells, and suppressing internal absorption loss can reduce the threshold current density of PCSELs. This will potentially lower heat generation within the device and further increases its maximum output power.

In 2021, the Noda research team at Kyoto University [40] expanded on previous efforts and produced a two-dimensional photonic crystal surface-emitting semiconductor laser with a continuous power output of 29 W at room temperature. The team improved the laser’s performance by increasing its gain through enlarging and optimizing the photonic crystal’s geometry. By adjusting the air-filling factor and the distance between two cells, the researchers were able to isolate the fundamental transverse mode and suppress all other resonances effectively. The fabricated laser has a working wavelength of 940 nm and delivers a divergence angle of less than 0.4°.

#### 3.2.3. Open-Dirac Cavities

In 2012, Bravo-Abad and colleagues [58] proposed using photonic Dirac cones to create large-area single-mode photonic crystal structures. They reported on an all-dielectric 3D photonic material with Dirac-like dispersion in a quasi-2D system that significantly enhances spontaneous emission coupling efficiency (β factor) across a large surface. This solution addresses the issue of rapid β factor degradation with an increase in system size, enabling photonic crystal large-area preparation.

In 2022, Kanté’s research group [59] proposed and demonstrated, through an experiment, an open Dirac cavity with linear dispersion. The use of a truncated photonic crystal arranged in a hexagonal pattern allowed the formation of an open Dirac cavity (Figure 19), suspended entirely in the air, and connected to the primary membrane using six bridges at the hexagon’s corners for mechanical stability. The cavity exhibits unique loss scaling in the reciprocal space. As a result, single-mode lasing remained constant as the cavity size was scaled, attributed to the special flat-envelope fundamental mode that locks all unit cells’ phasing within the cavity, thereby resulting in single-mode lasing. The researchers aptly named the solution Berkeley surface-emitting lasers (BerkSELs).

#### 3.2.4. Topological Cavity Structure

In 2020, Professor Ma Remnin’s team from the Institute of Physics, Chinese Academy of Sciences utilized a honeycomb photonic crystal structure embodied with a vortex Dirac mass [60]. The researchers improved the widely used one-dimensional feedback structures of industrial semiconductor lasers, namely phase-shifted DFB and VCSEL, to two-dimensional versions similar to the simulation of the Jackiw–Rossi zero mode. The topological cavity offers a single mid-gap mode with a continuous tunable mode diameter that ranges from a few microns to millimeter scale. Theoretically predicted and experimentally demonstrated, Dirac vortex cavities with tunable mode areas that span several orders of magnitude offer arbitrary mode degeneracy, robust large free spectral range, low-divergence vector beam output, and significantly high compatibility with high-refractive-index substrates. Consequently, the topological cavity facilitates stable single-mode operations of photonic crystal surface-emitting lasers (PCSELs).

In 2022, Yang Lechen et al. from the Institute of Physics, Chinese Academy of Sciences, performed a study [61] that involved the utilization of a Dirac-vortex topological cavity structure to create a topological cavity surface emitting laser (TCSEL), which achieved peak power of 10 W at a wavelength of 1550 nm, less than 1° divergence angle and 60 dB side mode suppression. TCSEL fabrication involves depositing a-Si on the MQW by chemical vapor deposition, and subsequently patterned the a-Si layer using electron beam lithography (EBL) and dry etching. Such procedures were clearly observed through optical microscopy and scanning electron microscopy (SEM). The vortex structure of the TCSEL can be seen in Figure 20.

### 3.3. Mirror-Type HCSEL

In 1989, Hamao, Sugimoto, and their colleagues [62] utilized 45-degree inclined reactive ion beam etching technology to produce a surface-emitting GaAs-AlGaAs laser featuring a 45-degree total reflection mirror, as depicted in Figure 21. The mirror angle error had an accuracy of 1°. Additionally, the power output ratio between surface-emitting and edge-emitting lights was remarkably high at 77%.

In 1991, the American Research Center [63] demonstrated the first ever GaAs/AlGaAs surface-emitting laser with a 45° internal micromirror. The device’s 45° and 90° mirrors were fabricated using ion beam etching and reactive ion beam etching techniques, respectively, to enhance their internal structure. The device utilizes internal micro-mirrors to reflect the beam horizontally towards the substrate. The surface of the laser emits light, and it was realized as shown in Figure 22. During quasi-continuous wave operation, the threshold current density was found to be 440 A/cm^2^ and the output power was achieved at more than 1 W.

In 2006, BinOptics Corporation [28] developed a 1300 nm high-power vertically emitting Fabry–Perot laser. The structure utilizes ridge waveguide technology and a chemically assisted ion beam-etching process. Additionally, they fabricated a 30 mW vertically emitting FP laser by fabricating a 45° dry-etched facet at one end of the laser cavity and a 90° etched facet at the other end. In addition, they integrated a laser and a monitor photodiode (MPD) in the same material. This simplifies the packaging process and reduces packaging and assembly costs. Additionally, the HCSEL with integrated MPD has passed more than 3000 h of reliability testing.

In 2013, the Central Research Laboratory of Hitachi Co., Ltd. in Tokyo, Japan, created an integrated photonic device that combined a monolithic lens-integrated SE-DFB laser array and p-i-n photodiode array [64]. The device fabrication process involved dry etching and wet chemical etching to produce monolithic InP lenses with a parabolic shape that facilitates the collimation of Gaussian beams. The lens-integrated surface-emitting DFB laser produces a circular, narrow far-field spot measuring 3.9° × 3.5°. With a coupling efficiency of 3 dB between the laser and single-mode fiber, the resulting four-channel lens-integrated DFB laser array operates as a 100 Gbps (25 Gbps per channel) optical transceiver even at high temperatures of 85 °C.

In 2014, researchers at the Central Research Laboratory of Hitachi Co., Ltd., in Tokyo, Japan, presented a lens-integrated surface-emitting laser (LISEL) incorporating an InGaAlAs DFB laser [65]. Using a silicon-on-insulator substrate, this new form of silicon photonic light source incorporated a grating coupler. The LISEL in this light source comprised a 45° rearview mirror and an integrated lens with a curvature radius of about 110 µm to produce a narrow far-field spot measuring 8.5° × 3.6°. Through the prediction of the alignment tolerance and spot size of the LISEL, researchers designed a suitable aperture size of the grating coupler. This design indicates that the excessive loss of the structure due to a misalignment of ±5 µm amounted to only 1.5 dB. This study demonstrated the feasibility of LISEL implementation on a passively aligned silicon platform and the functionality of this LISEL/grating coupler (GC) hybrid configuration light source in silicon photonic devices.

In 2015, Hitachi [66] achieved direct optical coupling between a lens-integrated surface-emitting DFB laser (LISEL) array and a single-mode fiber (SMF) array by monolithically integrating the InP lens and laser chip. The system achieved a small far-field angle of ~2° and an average maximum coupling efficiency of −8.4 dB on all four channels. Moreover, the measured misalignment tolerance between the LISEL array and the SMF array allows for a variance of ±5.5 µm, further lowering assembly costs and enabling the creation of smaller and more affordable 100 Gb/s optical modules.

In 2016, Hitachi Ltd. [67] continued their innovative streak by developing a cost-effective optical subassembly (OSA) that utilizes lens-integrated surface-emitting lasers (LISELs) for high-speed optical interconnection. The LISEL μ platform comprises the LISEL and its installation carrier, while the OSA on the silicon-photonics (SiP) platform consists of an external modulator, a grating coupler (GC), and a 3-dB coupler. This structure features low coupling losses, with the single-mode fiber having a loss of only 4.9 dB while the SiP platform has a loss of 4 dB. Additionally, experiments demonstrate that the structure does not require optical isolators while having the potential of non-hermetic packaging.

In 2019, Northwestern University [38] developed a reflective outcoupler that generates reflective outcoupling in a semiconductor waveguide using mask shape transfer to form the device’s mirrors. The system generates up to 6.7 W of single-mode peak power in the 90 cm^−1^ (215 nm) spectral range, exceeding 6 W of power. Furthermore, the system generates a high-quality output beam using a basic single-layer anti-reflection coating.

## 4. Summary and Prospects

Horizontal cavity surface-emitting lasers (HCSELs) are exemplary light sources for future applications due to their high output power and beam quality. In this paper, we provide an overview of the fundamental principles of three different HCSELs: second-order grating SE-DFB lasers, photonic crystal diffraction HCSELs, and mirror-type HCSELs. We also review the latest research progress on these three HCSELs. HCSELs have experienced rapid development over the last decade. Second-order grating SE-DFB lasers are capable of obtaining stable wavelengths and surface emissions with high beam quality, narrow spectral lines, and other significant advantages. Curved second-order grating SE-DFB lasers can reach a maximum continuous output power of 73 W. The development of high-order gratings and new materials, such as perovskite and organic semiconductor materials, is expected to further decrease manufacturing costs and threshold currents. Photonic crystal diffraction structures provide lasers with high output power, beam quality, and narrow divergence angles. The Noda team at Kyoto University proposed a large-area dual-lattice photonic crystal laser that delivered a continuous output power of 29 W and a divergence angle of less than 0.4°, surpassing the previous limitations of surface-emitting semiconductor lasers in high-power fields. Meanwhile, mirror-type HCSELs have simpler working principles, but their manufacturing processes are more complicated, and they have larger divergence angles, which have not been given as much attention. Nonetheless, improving the low cost and high beam quality of HCSELs remains a significant challenge for researchers. Especially for SE-DFB structures and photonic crystal diffraction structures, key factors for promoting further development are the expansion of the chip area, the improvement of surface coupling efficiency (such as buried second-order gratings), the suppression of high-order modes, and the increase in output power to broaden the laser’s applications in fields such as material processing, laser medicine, non-linear optics, and optical communication. In the future, a high-power level cavity laser light source with high beam quality and high slope efficiency is worth anticipating and will have vast potential in various industries.

## Figures and Tables

**Figure 1 sensors-23-05021-f001:**
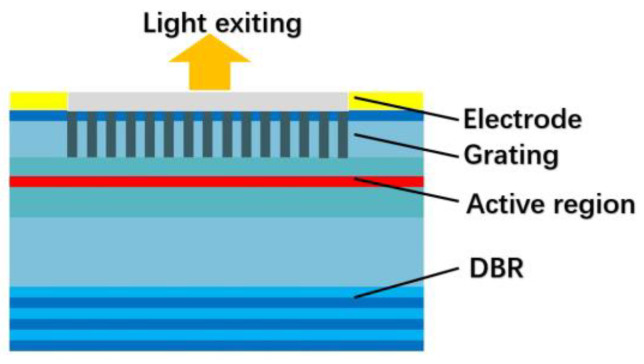
Scheme of the SE-DFB laser structure with the grating placed on the P surface (P surface-emission).

**Figure 2 sensors-23-05021-f002:**
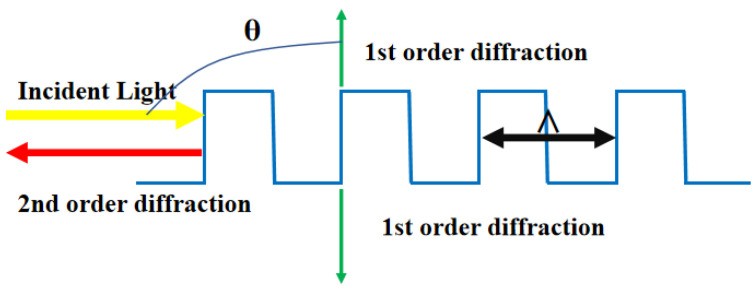
Diffraction characteristics of a second-order grating.

**Figure 4 sensors-23-05021-f004:**
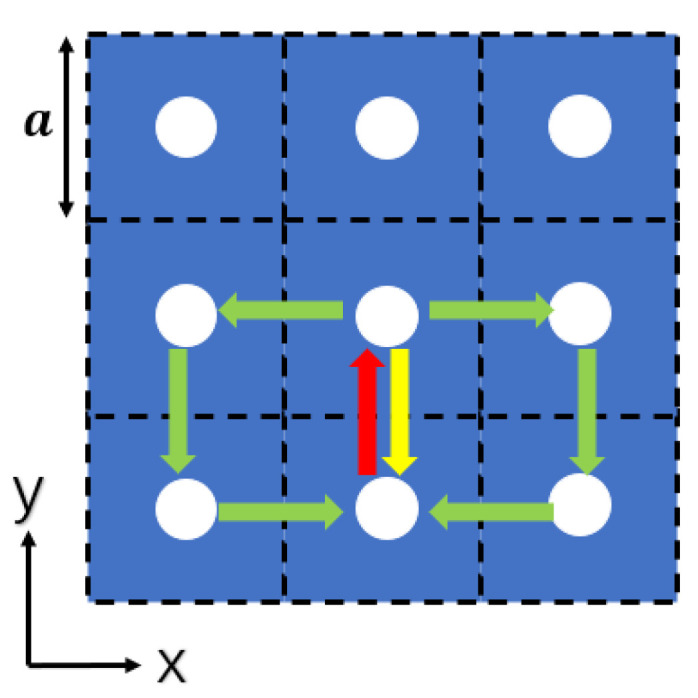
Scheme of light diffraction in photonic crystal in-plane. The red arrows represent light in the +y direction, while the green arrows represent corresponding, first-order Bragg diffracted light. Similarly, yellow arrows represent second-order Bragg diffracted light.

**Figure 5 sensors-23-05021-f005:**
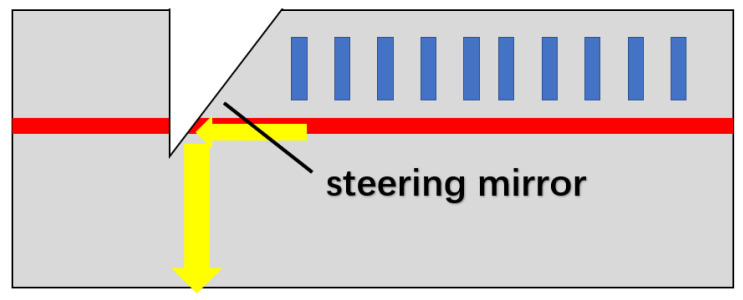
Scheme of mirror surface-emitting laser. The yellow arrows indicate the internal light path, where the light in the parallel direction is redirected towards surface emission by passing through the 45° reflector.

**Figure 6 sensors-23-05021-f006:**
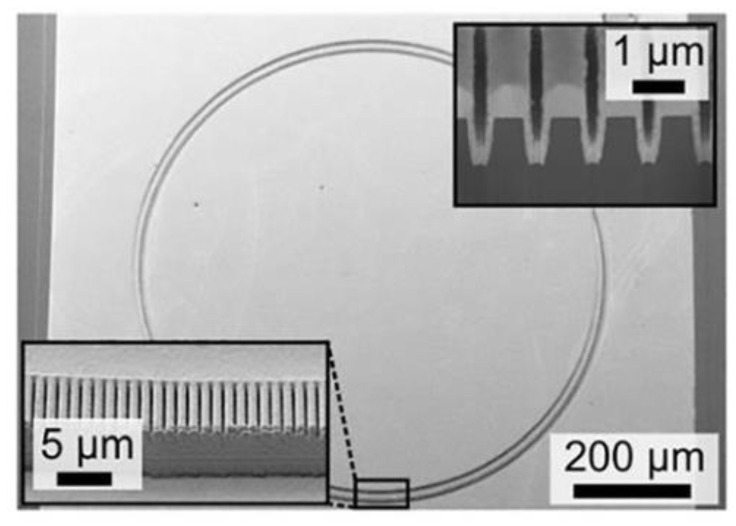
Scanning electron micrograph of an annular interband cascaded laser. The bottom left inset shows a metallized second-order DFB grating. © AIP Publishing. Copyright 2020 Applied Physics Letters.

**Figure 7 sensors-23-05021-f007:**
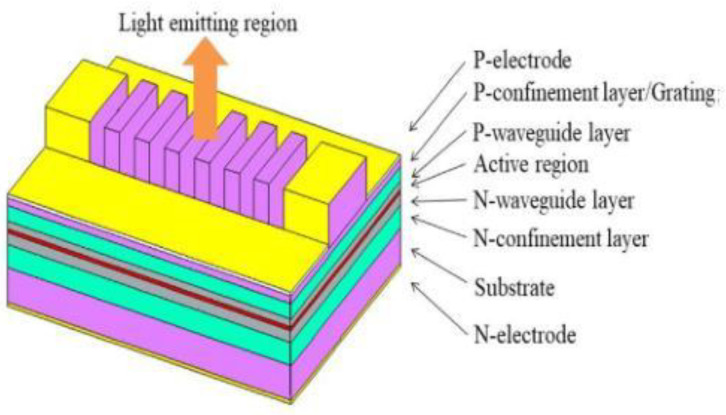
Three-dimensional structure diagram of SE-DFB laser. © Elsevier. Copyright 2021 Optics Communications.

**Figure 8 sensors-23-05021-f008:**
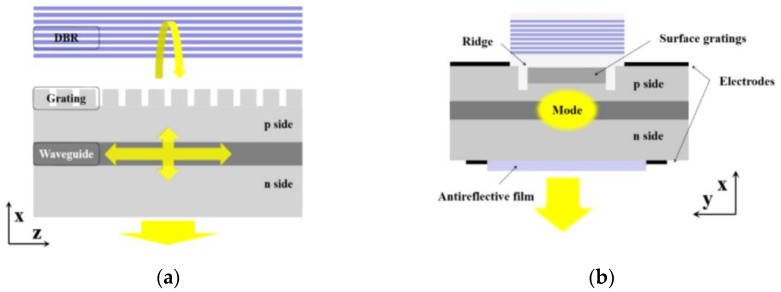
Scheme of grating-coupled surface-emitting laser and DBR. (**a**) Longitudinal section; (**b**) cross-section. © Optical Society of America. Copyright 2022 Optics Express.

**Figure 9 sensors-23-05021-f009:**
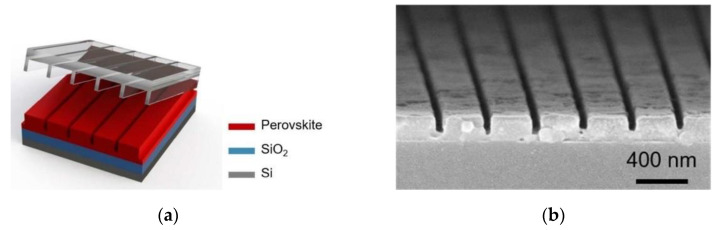
Scheme of the nanoimprint process for surface-emitting perovskite DFB lasers (**a**) and SEM images of imprinted MAPbI_3_ DFB gratings (**b**). © American Chemical Society. Copyright 2018 ACS Nano.

**Figure 10 sensors-23-05021-f010:**
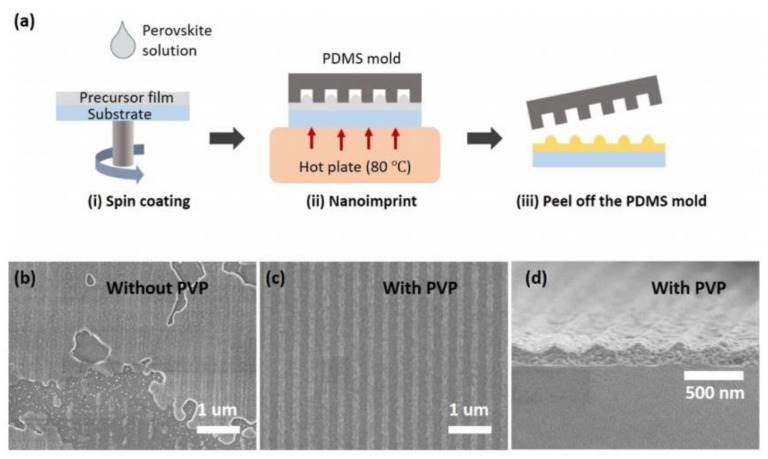
(**a**) Scheme of the process for fabricating a perovskite DFB cavity. Top-view SEM images of nanopatterned perovskite films without (**b**) and with (**c**) 20% polyvinylpyrrolidone (PVP). (**d**) Cross-sectional SEM image of the nanopatterned perovskite film with 20% PVP. © American Chemical Society. Copyright 2022 ACS Photonics.

**Figure 11 sensors-23-05021-f011:**
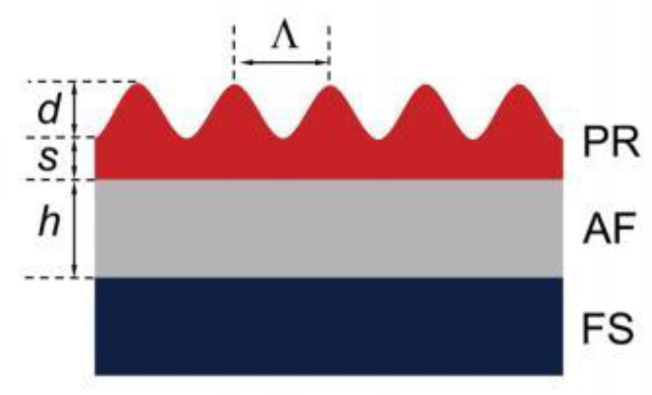
Scheme of the structure of an organic DFB laser, the grating consists of a photo-resist (PR) layer deposited on an active layer (AF) prepared on a fused silica (FS) substrate. © Springer Nature. Copyright 2019 Scientific Reports.

**Figure 12 sensors-23-05021-f012:**
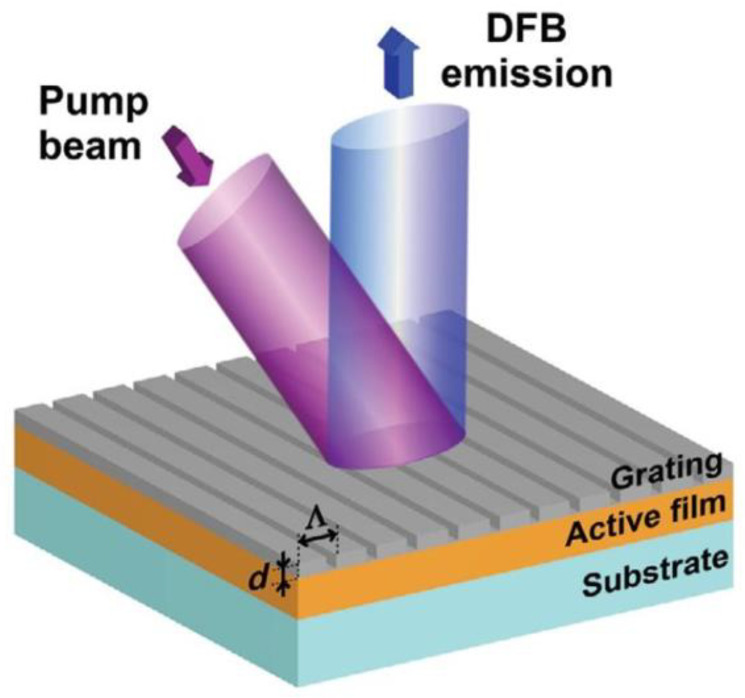
Scheme of a mixed-order DFB grating structure. © John Wiley and Sons. Copyright 2020 Advance Optical Materials.

**Figure 13 sensors-23-05021-f013:**
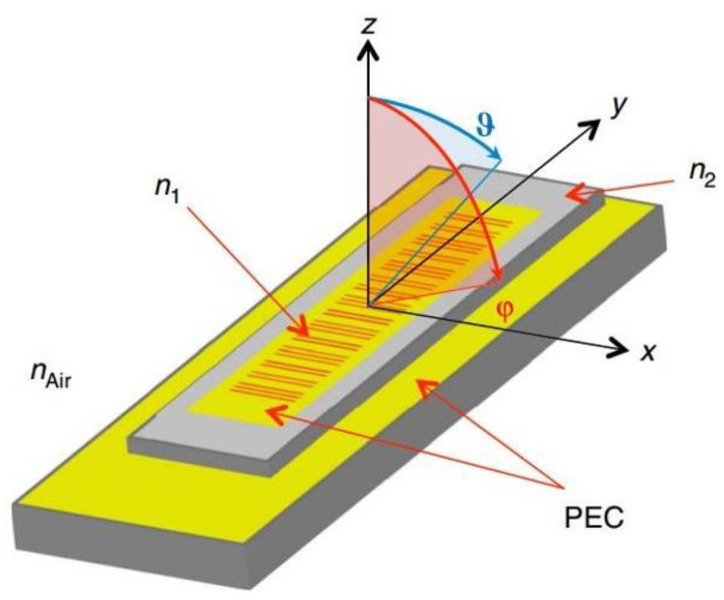
Scheme of the Octanacci surface grating laser. © Springer Nature. Copyright 2020 Light: Science & Applications.

**Figure 14 sensors-23-05021-f014:**
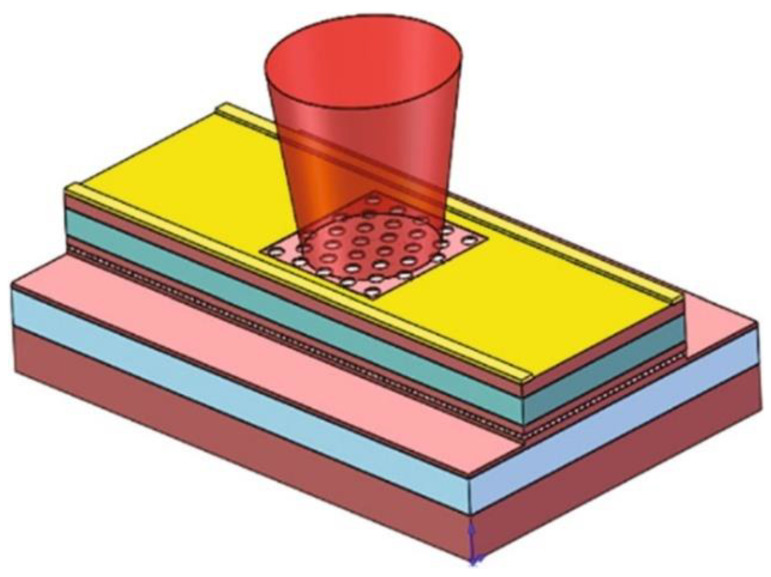
Scheme of the structure of 1.3 μm quantum dot PCSEL. © Springer Nature. Copyright 2019 Light: Science & Applications.

**Figure 15 sensors-23-05021-f015:**
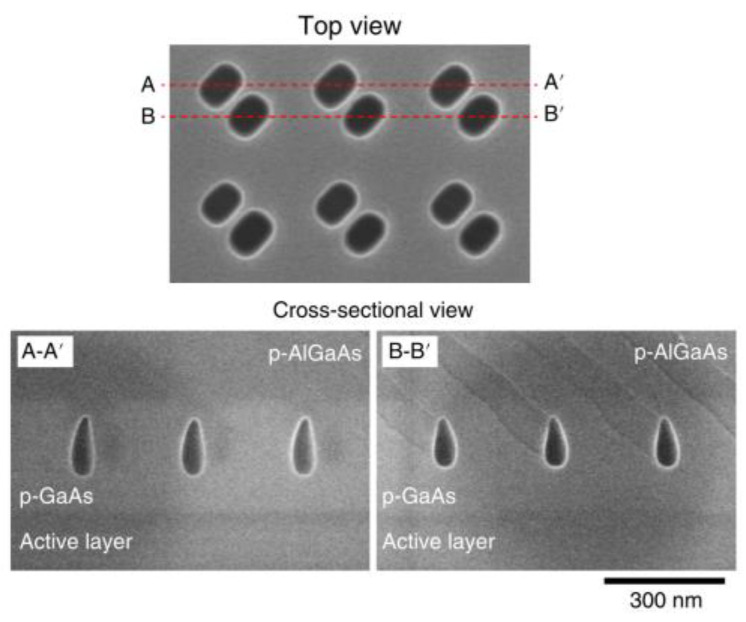
Top and cross-sectional SEM images of the double-lattice photonic crystal after etching. © Springer Nature. Copyright 2018 Nature Materials.

**Figure 16 sensors-23-05021-f016:**
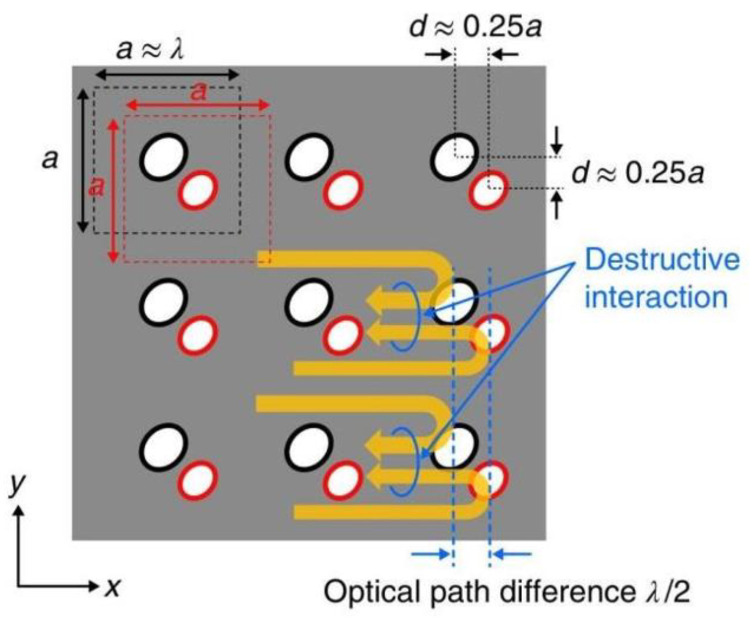
Schematic diagram of a double-lattice photonic crystal resonator, which consists of two lattice point groups. © Springer Nature. Copyright 2018 Nature Materials.

**Figure 17 sensors-23-05021-f017:**
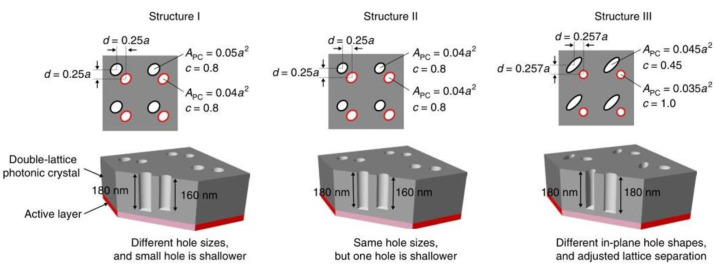
Top and cross-sectional views of three types of double-lattice photonic crystals. © Springer Nature. Copyright 2018 Nature Materials.

**Figure 18 sensors-23-05021-f018:**
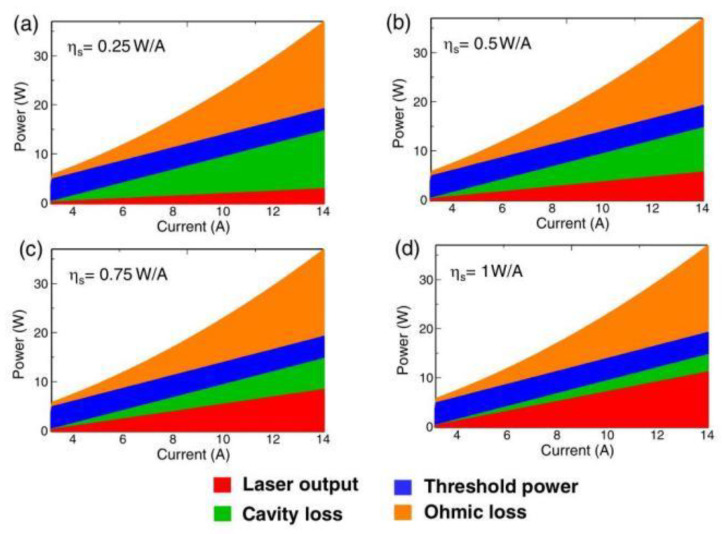
(**a**–**d**) Calculated power consumption of 800-µm-diameter PCSELs as a function of injection current above the threshold, for slope efficiencies ranging from 0.25 to 1 W/A. © Optical Society of America. Copyright 2020 Journal of the Optical Society of America B.

**Figure 19 sensors-23-05021-f019:**
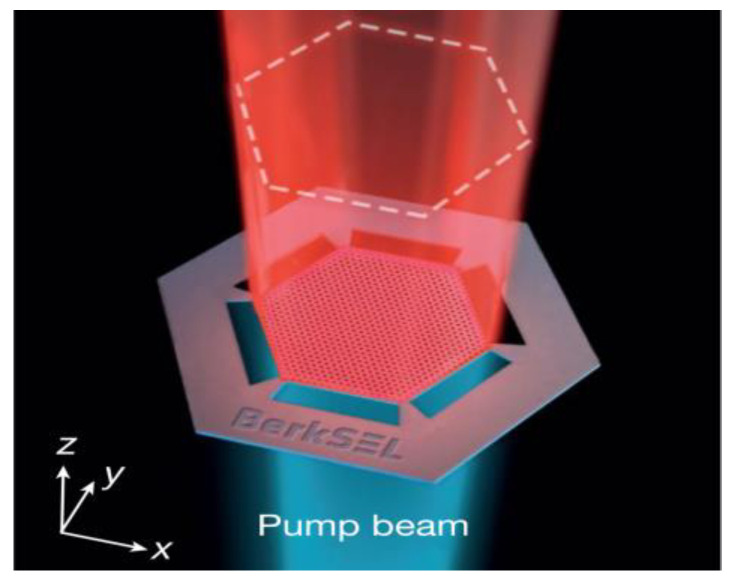
Berkeley surface-emitting laser structure. © Springer Nature. Copyright 2022 Nature.

**Figure 20 sensors-23-05021-f020:**
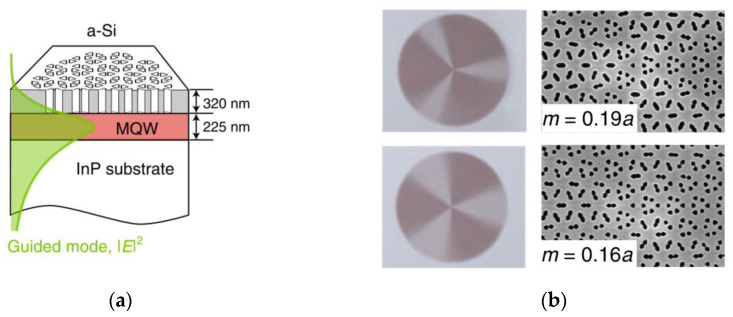
(**a**) Schematic diagram of topological cavity surface-emitting laser (TCSEL) structure; (**b**) optical microscope and SEM images of TCSELs. © Springer Nature. Copyright 2022 Nature Photonics.

**Figure 21 sensors-23-05021-f021:**
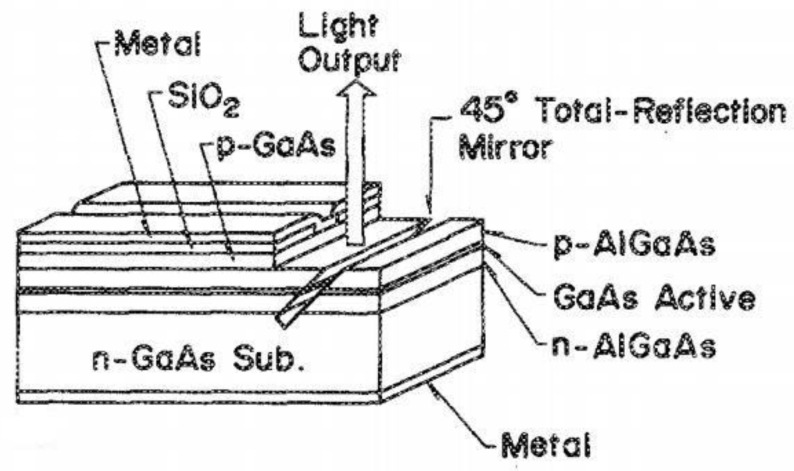
Device structure (reactive ion beam etching). © AIP Publish. Copyright 1989 Applied Physics Letters.

**Figure 22 sensors-23-05021-f022:**
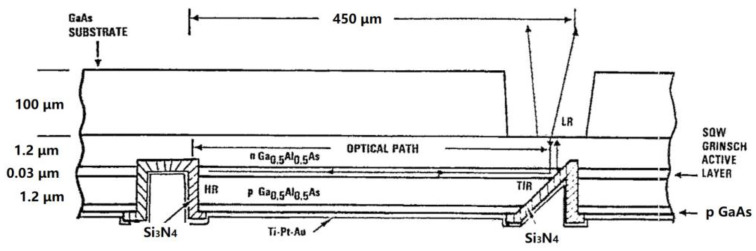
Scheme of a surface-emitting laser with internal 45° and 90° micromirrors. © AIP Publish. Copyright 1991 Applied Physics Letters.

**Table 1 sensors-23-05021-t001:** Recent achievements in HCSELs.

Year	Institution	Band	Output Power	Slope Efficiency	Divergence Angle	Structure	Ref
2005	University of Wisconsin	980 nm	20 units form an array: 1.6 W (continuous wave (C.W.))			Linear second-order grating	[29]
2006	BinOptics Corporation	1.3 μm	30 mW	~0.3 W/A	15° × 36°	mirror-type HCSEL	[28]
2012	Alfalight Company	97x nm	68 W (C.W.)	0.8 W/A	<8°	curved second-order grating	[30]
2014	Kyoto University	941 nm	1.5 W (C.W.)	0.66 W/A	<3°	Photonic crystals	[31]
2016	Taiwan National Chiao Tung University	1.3 μm	2 mW		1° × 8~9°	Linear second-order grating	[32]
2016	Institute of Semiconductors of the Chinese Academy of Sciences	7 μm	2.29 W	500 mW/A	2.9° × 0.36°	Linear second-order grating	[33]
2017	Institute of Semiconductors of the Chinese Academy of Sciences	4.97 μm	248 mW		0.14° × 16°	Linear second-order grating	[34]
2018	Lehigh University	3.4 THz	170 mW	993 mW/A	5° × 25°	Linear high-order grating	[35]
2018	Kyoto University	940 nm	7 W (C.W.)	0.48 W/A	<0.4°	Double lattice	[36]
2019	Changchun Institute of Optics, Fine Mechanics and Physics	1.3 μm	13.3 mW (C.W.)	40.9 mW/A		Quantum-dot Photonic-crystals	[37]
2019	Northwestern University	4.9 μm	6.7 W (peak power)			mirror-type HCSEL	[38]
2021	Changchun University of Science and Technology	976 nm	84 mW		2.6° × 6.1°	Linear second-order grating	[39]
2021	Kyoto University	940 nm	29 W (C.W.)	~0.66 W/A	<0.4°	Double lattice	[40]

## Data Availability

Not applicable.

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
