# Peer review of "Research Progress of Horizontal Cavity Surface-Emitting Laser"

_sensors, 2023, doi:10.3390/s23115021_

Round 1
Reviewer 1 Report
Please see attached.

There are typos in the manuscript, please check the review comments.
Author Response
Thank you very much for approving our work. Your comments are very valuable and helpful for improving our paper, as well as the important guiding significance to our future research. We have revised the manuscript according to your comments. Our point-by-point response is attached below.
Point 1: Comments regarding to the definition of photonic crystal surface emitting laser (PCSEL): I am a little confused with the relationship between PCSEL, VCSEL and HCSEL, apparently there are several groups used photonic crystal pattern on vertical cavity surface emitting laser (VCSEL) to control lateral modes.
Response 1: Thank you for your constructive and useful advice. I noticed my informal and loose use of the word. Photonic crystal is used in both VCSEL and HCSEL. The PCSELs cited in this article are different from VCSELs, surface emission is realized by PC diffraction. In order to prevent misunderstanding, we have changed the title of PCSELs to photonic crystal diffraction structure. At the same time, relevant inappropriate places have been modified. (Page 3, Page 13 and Page 19)
Point 2: Typos on page 15 and page 16, “3.2.4 Topological cavity structure” appeared twice, please delete the first one.
Response 2: Thank you for reminding us. We have deleted the first one. (Page 16)
Point 3: Besides the paper listed by the author, there are several important papers of HCSEL should be cited and reviewed as well, please consider papers below:
(i) “Horizontal cavity vertically emitting lasers with integrated monitor photodiodes” Proceedings Volume 6352, Optoelectronic Materials and Devices; 63520U (2006) https://doi.org/10.1117/12.689153
(ii) “Photonic crystal vertical-cavity surface-emitting lasers with true photonic bandgap” https://opg.optica.org/ol/abstract.cfm?uri=ol-35-6-82
Response 3: Thank you for constructive and useful advice. I am very sorry for the omission of these documents. I have done some serious reading of these excellent literatures and have included them in the article. The specific content in the manuscript is as follows: (Page 3 and Page 18)
(i) “In 2006, BinOptics Corporation [28] introduced a 1300nm high-power vertically emitting Fabry-Perot laser that uses ridge waveguide technology and a chemically assisted ion beam etching process. Moreover, a 30mW vertically emitting Fabry-Perot laser was fabricated with a 45° dry-etched facet at one end of the laser cavity and a 90° etched facet at the other end. BinOptics also managed to simplify the packaging process and reduce assembly costs by integrating a laser and a monitor photodiode (MPD) via the integration of a rear monitor diode in the same material. This integration was critical as the HCSEL, including the MPD component, had passed over 3000 hours of reliability testing.”
(ii) “Photonic crystal surface emitting lasers (PCSELs) can be classified as either horizontal cavity structures or vertical cavity structures [22] based on cavity design. The horizontal cavity surface emitting lasers (HCSELs) utilize the photonic crystal diffraction of the photonic crystal band-edge mode and rely on the band-edge resonance effect of the two-dimensional photonic crystal to produce stimulated emission amplification and surface emission. The photonic crystal (PC) can be viewed as a two-dimensional grating structure, similar to the lasing principle of surface emitting distributed feedback (DFB) lasers.”

Reviewer 2 Report
The paper revise in a very broad sense the various geometries proposed in the field of semiconductor lasers to achieve a surface emission from a longitudinal resonant cavity. It might be of help for the laser community.
Acceptable.
Author Response
Thank you very much for approving our work. Your comments are very valuable and helpful for improving our paper, as well as the important guiding significance to our future research. As per your suggestions, we have revised the manuscript and made the necessary changes to refine its language and structure. Particularly, we have concentrated on polishing the language in the manuscript to ensure it meets academic standards. Additionally, we have streamlined the structure of the article by removing any obscure and redundant parts. Once again, we thank you for your meticulous evaluation, which has proven to be a valuable contribution to our work.

Reviewer 3 Report
The paper provides a comprehensive review of the construction of a group of semiconductor lasers known as horizontal cavity surface emitting lasers (HCSELs). However, it has some basic shortcomings that could be improved. Firstly, it lacks tables, charts, etc. that could help in summarizing the content, and secondly, it lacks deeper analysis and thoughts that could enhance the review. Nevertheless, the paper is still worth publishing.
Here are some specific issues that need to be corrected:
- The introduction of the paper suggests that the goal is always to obtain the highest possible power emitted from semiconductor lasers. However, this is not entirely true. In many applications, such as telecommunications, the aim is to limit the electrical power supplied to the device and to use it effectively.
- Citation [26] seems to be misquoted.
- The text lacks reference to Figure 8.
- In line 445, "M squared (M_sup_2)" should be used instead of "M2".
- In line 448, “200mm x 200mm”. This means that it is probably a mistake. Probably the authors meant the size of a single element of the photonic crystal (triangle) which is 200 nm x 200 nm. The authors should clarify what they mean.
- The text lacks reference to Figure 20.
- In line 593, "CaAS/AlGaAs" should be replaced with "GaAs/AlGaAs".
- Citations [62] in the References need to be corrected.
Author Response
Thank you very much for approving our work. Your suggestions are very valuable and helpful for improving our paper. We have concentrated on polishing the language in the manuscript to ensure it meets academic standards. And we have revised the manuscript according to your comments. Our point-by-point response is attached below.
Point 1: This manuscript lacks tables, charts, etc. that could help in summarizing the content The introduction of the paper suggests that the goal is always to obtain the highest possible power emitted from semiconductor lasers. However, this is not entirely true. In many applications, such as telecommunications, the aim is to limit the electrical power supplied to the device and to use it effectively.
Response 1: Thank you for your constructive and useful advice. We have included the appropriate amount of tables in the article (Table 1). The details are as follows: (Page 5)
Table 1. Recent achievements in HCSELs.
Year |
Institution |
Band |
Output Power |
Slope Efficiency |
Divergence Angle |
Structure |
Ref |
2005 |
University of Wisconsin |
980 nm |
20 units form an array: 1.6 W (continuous wave) |
|
|
Linear second order grating |
[29] |
2006 |
BinOptics Corporation |
1.3 μm |
30 mW |
~0.3 W/A |
15°× 36° |
mirror-type HCSEL |
[28] |
2012 |
Alfalight Company |
97x nm |
68 W(C.W.) |
0.8 W/A |
<8° |
curved second order grating |
[30] |
2014 |
Kyoto University |
941 nm |
1.5 W(C.W.) |
0.66 W/A |
<3° |
Photonic crystals |
[31] |
2016 |
Taiwan National Chiao Tung University |
1.3 μm |
2 mW |
|
1°×8~9° |
Linear second order grating |
[32] |
2016 |
Institute of Semiconductors of the Chinese Academy of Sciences |
7 μm |
2.29 W |
500 mW/A |
2.9°×0.36° |
Linear second order grating |
[33] |
2017 |
Institute of Semiconductors of the Chinese Academy of Sciences |
4.97 μm |
248 mW |
|
0.14°× 16° |
Linear second order grating |
[34] |
2018 |
Lehigh University |
3.4 THz |
170 mW |
993 mW/A |
5°× 25° |
Linear high-order grating |
[35] |
2018 |
Kyoto University |
940nm |
7W(C.W.) |
0.48 W/A |
<0.4° |
Double-lattice |
[36] |
2019 |
Changchun Institute of Optics, Fine Mechanics and Physics |
1.3μm |
13.3 mW(C.W.) |
40.9 mW/A |
|
Quantum-dot Photonic-crystals |
[37] |
2019 |
Northwestern University |
4.9 μm |
6.7W(peak power) |
|
|
mirror-type HCSEL |
[38] |
2021 |
Changchun University of Science and Technology |
976nm |
84mW |
|
2.6°×6.1° |
Linear second order grating |
[39] |
2021 |
Kyoto University |
940nm |
29W(C.W.) |
~0.66 W/A |
<0.4° |
Double-lattice |
[40] |
Point 2: This manuscript lacks deeper analysis and thoughts that could enhance the review.
Response 2: Thank you for your constructive and useful advice. We have revised the Summary section in the manuscript. The details are as follows: (Page 21)
“Horizontal cavity surface-emitting lasers (HCSELs) are exemplary light sources for future applications due to their high output power and beam quality. In this paper, we provide an overview of the fundamental principles of three different HCSELs: second-order grating Surface Emission Distributed Feedback (SE-DFB) lasers, Photonic Crystal Diffraction HCSELs, and Mirror-type HCSELs. We also review the latest research progress on these three HCSELs. HCSELs have experienced rapid development over the last decade. Second-order grating SE-DFB lasers are capable of obtaining stable wavelengths and surface emissions with high beam quality, narrow spectral lines, and other significant advantages. Curved second-order grating SE-DFB lasers can reach a maximum continuous output power of 73 watts. The development of high-order gratings and new materials like perovskite and organic semiconductor materials is expected to further decrease manufacturing costs and threshold currents. Photonic Crystal Diffraction structures provide lasers with high output power, beam quality, and narrow divergence angles. For instance, the Noda team at Kyoto University developed a large-area dual-lattice photonic crystal laser that delivered a continuous output power of 29 W and a divergence angle of less than 0.4°, surpassing previous limitations of surface-emitting semiconductor lasers in high-power fields. Meanwhile, Mirror-type HCSELs have simpler working principles, but their manufacturing processes are more complicated, and they have larger divergence angles which have not been given as much attention. Nonetheless, improving the low cost and high beam quality of HCSELs remains a significant challenge for researchers. Especially for SE-DFB structures and photonic crystal diffraction structures, key factors to promote further development are the expansion of chip area, the improvement of surface coupling efficiency (such as buried second-order gratings), the suppression of high-order modes, and the increase of output power to broaden the laser's applications in fields such as material processing, laser medicine, nonlinear optics and optical communication. In the future, a high-power level cavity laser light source with high beam quality and high slope efficiency is worth anticipating and will have vast potential in various industries.”
Point 3: The introduction of the paper suggests that the goal is always to obtain the highest possible power emitted from semiconductor lasers. However, this is not entirely true. In many applications, such as telecommunications, the aim is to limit the electrical power supplied to the device and to use it effectively.
Response 3: Thank you for your constructive and useful advice. We have revised the Introduction section in the manuscript. The details are as follows: (Page1 and Page2)
“Semiconductor lasers have garnered significant interest among researchers due to their advantages, such as small size, long lifespan, high conversion efficiency, fast modulation rate, wide wavelength range, and ease of integration [1-2]. Over the course of their development, semiconductor lasers have found extensive use in various fields, including industrial production, military defense, and medical treatment [3-6]. Furthermore, they have emerged as crucial light sources in optical communication, optical pump laser, and optical information storage [6]. With the progression of society and the expansion of semiconductor laser application in laser radar, laser processing, and high-speed laser communication, individuals require laser light sources with lower divergence angles, higher power and higher slope efficiency [7-8]. Nonetheless, conventional semiconductor lasers have drawbacks such as susceptibility to cavity surface damage [9], significant divergence, and inadequate monochromaticity [10-11]. These conventional semiconductor lasers require complex systems for beam shaping, collimation, and coupling to attain high beam quality, but their high cost renders them inadequate for affordable use.
Commercial vertical-cavity surface-emitting semiconductor lasers (VCSELs) have superior performance with excellent beam shape, no cavity surface catastrophe damage, and easy two-dimensional integration [12]. However, the thin active region of VCSEL results in a lower single-way gain [13], which limits its output power severely. Even with lateral multimode, the competition from higher-order modes can degrade beam quality substantially [14, 15].
Horizontal-cavity surface-emitting semiconductor lasers (HCSELs) provide the advantages of high power and high slope efficiency [16]. The structure of HCSEL introduces a diffraction or reflection structure that improves the beam quality so that the light is emitted in a direction perpendicular to the epitaxial surface of the crystal. Realization of longitudinal mode characteristics of the surface-emitting laser is limited by the diffraction structure, which in turn has the advantages of small temperature drift and narrow spectrum [17]. Moreover, the power-bearing capacity of crystal epitaxial surface is higher than dissociation cavity surface, making it possible to withstand higher single-mode output power [18]. HCSEL also offers benefits such as high surface damage threshold, simple manufacturing, and easier 2D array integration [19, 20]. Therefore, this paper briefly describes the working principle, device structure, research progress, and development trends of three different HCSELs.”
Point 4: Citation [26] seems to be misquoted.
Response 4: Thank you for reminding us. We noticed that the cited article is mainly about the application of Steering mirror-type HCSEL in the field of optical communication and optical interconnection. Rather than an introduction based on laser structure. There is an inappropriate citation. Therefore, we have replaced the reference with: “Horizontal cavity vertically emitting lasers with integrated monitor photodiodes [27]” Proceedings Volume 6352, Optoelectronic Materials and Devices; 63520U (2006) https://doi.org/10.1117/12.689153. (Page 4, line 154)
Point 5: The text lacks reference to Figure 8.
Response 5: Thank you for reminding us. We have referenced Figure 8. (Page 9, line 338)
Point 6: In line 445, "M squared (M_sup_2)" should be used instead of "M2".
Response 6: Thank you for reminding us. We have replaced "M squared (M_sup_2)" with "M2". (line 524)
Point 7: In line 448, “200mm x 200mm”. This means that it is probably a mistake. Probably the authors meant the size of a single element of the photonic crystal (triangle) which is 200 nm x 200 nm. The authors should clarify what they mean.
Response 7: Thank you for reminding us. Sorry for not being clear in the manuscript. "200mm x 200mm" represents the photonic crystal chip size. I have corrected this mistake in the article. (line 527)
Point 8: The text lacks reference to Figure 20.
Response 8: Thank you for reminding us. We have removed Figure 20. It is repeated with the content of Figure 21. (Page 17)
Point 9: In line 593, "CaAs/GaAlAs" should be replaced with "GaAs/AlGaAs".
Response 9: Thank you for your suggestion. We have replaced " CaAs/AlGaAs " with " GaAs/AlGaAs ". (line 688)
Point 10: Citations [62] in the References need to be corrected.
Response 10: Thank you for reminding us. We have corrected the citation formatting. (Page 23, line 744 [38]) Due to revisions to the article, the citation is now Citations [38].

Reviewer 4 Report
Attached please find my comments.

It is included in the attached file.
Author Response
Thank you very much. Your comments are very valuable and helpful for improving our paper, as well as the important guiding significance to our future research. We have concentrated on polishing the language in the manuscript to ensure it meets academic standards. And we have revised the manuscript according to your comments. Our point-by-point response is attached below.
Section1: something misleading.
- a) In line 251-258, Ref. [35] in 2018 reported a THz quantum cascade lasers based on metal-metal waveguides. However, I find a similar paper in CLEO 2007 entitled “Single-Mode Surface-Emitting Terahertz Quantum-Cascade Lasers Operating up to ~ 150 K”. The authors should not ignore this earlier work and have to point out the advancement made over ten years.
Response a): Thank you for constructive and useful advice. I am very sorry for the omission of this document. I have done some serious reading of this excellent literature and have included it in the article. The specific content in the manuscript is as follows: (Page 6)
“In 2007, Sushil Kumar et al. [41] from the Massachusetts Institute of Technology reported a surface emission distributed feedback Terahertz quantum cascade single-mode laser, and obtained robust single-mode operation and single-lobe beam mode in this structure. They used a second-order distributed feedback grating to achieve surface coupling out of the laser beam. And they combined a metal-metal DFB structure with a resonant phonon THz QCL active region. With this structure, not only the low waveguide loss advantage of the metal-metal waveguide can be maintained, but also its output power can be increased and the beam quality can be improved. The frequency range of the quantum cascade single-mode laser is 0.35THz ~ 2.9THz. The continuous wave power of the laser reaches 6mW at 5K. And the maximum pulse working temperature can reach 149K.”
- b) In line 424-426, “Their two research groups …, bonding the structure …” is misleading. Actually, wafer bonding was performed by Imada’s group but not Meier’s.
Response b): Thanks a lot for your guidance on this. We have corrected “Their two research groups” to “Imada’s group”. (line 509-511)
- c) Simulated results should be clearly distinguished from experimental results to avoid misunderstanding. For example, line 227-238, line270-288 and line 324-334 are from simulations; however, their descriptions can be mistaken as from experiments.
Response c): Thank you for your constructive and useful advice. I am very sorry that this content caused your misunderstanding. We have made it clear in the manuscript that the experimental results are from simulations. (Page 7, Page 9 and Page 11)
- d) In line 239-247, “p-plane” and “n-plane” are not common expression. Use “p-surface” and “n-substrate” instead.
Response d): Thank you for your advice. We have replaced “p-plane” and “n-plane” with “p-surface” and “n-substrate” respectively. (line 302-305)
Section2: grammar issues
- a) In line 63-64, “… higher than that of dissociation cavity surface …” is somewhat problematic.
Response a): Thank you for reminding us. We have changed this sentence to: “Moreover, the power-bearing capacity of crystal epitaxial surface is higher than dissociation cavity surface, making it possible to withstand higher single-mode output power”. (line 268-270)
- b) In line 80-81, we have neither verb nor object in the sentence.
Response b): Thank you for reminding us. We have changed this sentence to: “A distributed feedback (DFB) grating structure is introduced to diffract the laser light from the horizontal to the vertical direction, achieving surface lasing”. (line 78-79)
- c) In line 350-352, “… organic and thin-film materials and the insertion …” is lengthy. I am afraid it cannot convey the idea
Response c): Thank you for reminding us. We have changed this sentence to: “This research represents a significant step forward for the development of electrically pumped lasers in thin-film and organic materials and for the integration of perovskite lasers into photonic circuits”. (line 410-412)
- d) In line 517-522, the sentence is too long and hard to read.
Response d): Thank you for reminding us. We have changed this sentence to: “The results indicate that a reduction in the slope efficiency increases the contribution of the resonator to the power consumption. This observation implies that by reducing internal absorption loss, optimizing the radiation constant. And tuning the number of quantum wells, the threshold current density of PCSELs can be diminished, thus minimizing the heat generated inside the device and increasing its maximum output power”. (line 609-614)
- e) In line 643-644, the sentence is strange and suspicious. The digit numbers of 4.9, 90, 215, 6.7 and 6 are absent in reference [65]. I am afraid it is wrongly cited.
Response e): Thank you for constructive and useful advice. We have changed this sentence to: “This structure features low coupling losses, with the single-mode fiber having a loss of only 4.9dB while the SiP platform has a loss of 4dB”. Due to personal oversight, reference [67] was miscited. We are very sorry. We have changed the reference to “Surface Emitting, Tunable, Mid-Infrared Laser with High Output Power and Stable Output Beam”. (line 738-740)
Section4: redundant, typographic and erroneous descriptions
- Redundant:
- The sentence in line 45-47 is almost the same as that in line 43-45.
Response a): Thank you for your advice. We have changed the sentences to: “These conventional semiconductor lasers require complex systems for beam shaping, collimation, and coupling to attain high beam quality, but their high cost renders them inadequate for affordable use.” (line 45-46)
- In line 345, the word “perovskite” seems redundant.
Response b): Thank you for reminding us. We modified the paragraph. The modified content is “In a recent study, the University of Texas at Dallas reported the successful fabrication of surface-emitting distributed feedback methylammonium lead iodide (MAPbI3) perovskite lasers on silicon substrates. The perovskite film was patterned using thermal nanoimprint lithography (NIL) to achieve predefined cavity geometries, size control, repeatability, and high-Q-factor cavities with large mode gain overlap (Figure 9). This technique also improved the material's emission characteristics. The perovskite laser demonstrated continuous laser output at room temperature and an ultra-low pump power density of 13 W/cm2. This research represents a significant step forward for the development of electrically pumped lasers in thin-film and organic materials and for the integration of perovskite lasers into photonic circuits.” (line 403-408)
- In line 379, it is possible to realize the realization of ……
Response c): Thank you for reminding us. We have changed the sentence to: “Accordingly, this research takes a significant step towards the development of genuine continuous wave organic laser technology, creating possibilities to realize high-power surface-emitting organic semiconductor lasers on a large scale in the future.” (line 452-455)
- In line 467, “as much as” seems redundant.
Response d): Thank you for reminding us. We have changed the sentence to: “The dual-lattice structure was designed to enhance the vertical light output by minimizing resonances, except for the fundamental transverse mode, through optimization of both the air filling factor and grid distance.” (line 550-552)
- In line 468, The device PC and MQW layers ……
Response e): Thank you for reminding us. We have changed the sentence to: “The dual-lattice structure comprises cladding layers, sandwiching the PC and MQW layers, with p-type and n-type conductivity. Successful fabrication of the dual-lattice structure was performed using MOVPE crystal growth and high-precision collimated electron beam lithography, in conjunction with two-step dry etching methods.” (line 558-562)
- In line 483-484, “… l/2 to make in-plane destructive …” is repeated again in line 492-493.
Response f): Thank you for your advice. We have modified the following sentences. We have changed the sentence to: “The double-lattice photonic crystal resonator operated on the principle of misaligning the two lattice point groups by λ/4 in both the horizontal and vertical directions.” (line 581-583)
- The sentence in line 495-498 is almost the same as that in line 484-486.
Response g): Thank you for your advice. We have removed the redundant part of the following. (line 588-591)
- In line 526, “3.2.4 Topological cavity structure” is misprinted.
Response h): Thank you for your advice. We have removed the redundant part. (Page 16, line 622)
- i) In line 577, the phrase “surface structure” seems redundant.
Response i): Thank you for your advice. We have removed the redundant part. (line 670)
- Typo:
- a) In line 57-58, “… the advantages of both: high power and high efficiency” is ambiguous in context. What do you mean, slope efficiency or coupling efficiency?
Response a): Thank you for your advice. Sorry for the misunderstanding, the content of this position is “slope efficiency”. We have made corrections in the manuscript. (line 60)
- b) In Figure 2 of page 3, “2st” order diffracted light should be “2nd”.
Response b): Thank you for reminding us. We have modified Figure 2. The modified picture is as follows: (Page 3)
- c) In line 162-163, “… etching process … is relatively high” is ambiguous in context.
Response c): Thank you for your advice. We have changed the sentence to: “Thus, the etching process of this structure on the steering mirror is significant.” (line 162-163)
- d) In line 287, the word “conductive” is ambiguous.
Response d): Thank you for your advice. We have changed the sentence to: “Furthermore, its transparency yields low-loss vertical light output while still providing excellent ohmic contact. In addition, the relatively low refractive index of ITO provides good light confinement.” (line 215-217)
- e) In line 411, “Octanacci” should be corrected as “Octonacci sequence”.
Response e): Thank you for reminding us. We have corrected “Octanacci” to “Octonacci sequence”. (line 489)
- f) In line 433, “g” should be corrected as “G”.
Response f): Thank you for your advice. We have corrected “g-point” to “G-point”. (line 511)
- g) In line 458-459, This study provides …… and “expands” ……
Response h): Thank you for reminding us. We have corrected “expand” to “expands”. (line 541)
- In line 653, “… spectral lines. Wide, good …” is badly punctuated.
Response i): Thank you for reminding us. We have changed the sentence to: “Second-order grating SE-DFB lasers are capable of obtaining stable wavelengths and surface emissions with high beam quality, narrow spectral lines, and other significant advantages.” (line 757-759)
- Erroneous:
- a) In line 66, “small surface damage threshold” means easily damaged, which is a disadvantage.
Response a): Thank you for reminding us. We have corrected " small surface damage threshold " to " high surface damage threshold ". (line 69)
- b) In line 270, Tian “Kun” instead of Tian “Yu”.
Response b): Thank you for your advice. We have corrected " Yu " to " Kun ". (line 330)
- c) In line 276, it has nothing to do with “silicon” wafer.
Response c): Thank you for reminding us. We have corrected " silicon wafer " to " wafer " (line 335)
- d) In line 448, “200mm´200mm” should be corrected as “200mm´200mm”.
Response d): Thank you for reminding us. We have corrected “200mm´200mm” to “200mm´200mm”. (line 527)
- e) In line 467, it is more precise to use “two lattice points” instead of “two grids”.
Response e): Thank you for reminding us. We have corrected “two grids” to “two lattice point”. (line 582)
- f) In line 499, k1D are diffraction “coefficients” instead of “intensities”.
Response f): Thank you for reminding us. We have corrected " coefficients " to " intensities " (line 592)
- g) In line 511, … 300 mm thick “submount” instead of “base” bonded to ……
Response g): Thank you for reminding us. We have corrected " base " to " submount ". (line 605)
- h) In line 579, “electron microscopy” should be added after scanning and ended with punctuation.
Response h): Thank you for your advice. We have corrected " scanning " to " electron microscopy ". And we have filled in the missing punctuation. (line 673)
Section4: line 650-668 in the fourth section of summary and prospects is too random and can be omitted in my opinion
Response section4: Thank you for your constructive and useful advice. W We have carefully revised the summary section. After modification and refining, the content is as follows: (Page 21 and Page 22)
" Horizontal cavity surface-emitting lasers (HCSELs) are exemplary light sources for future applications due to their high output power and beam quality. In this paper, we provide an overview of the fundamental principles of three different HCSELs: second-order grating Surface Emission Distributed Feedback (SE-DFB) lasers, Photonic Crystal Diffraction HCSELs, and Mirror-type HCSELs. We also review the latest research progress on these three HCSELs. HCSELs have experienced rapid development over the last decade. Second-order grating SE-DFB lasers are capable of obtaining stable wavelengths and surface emissions with high beam quality, narrow spectral lines, and other significant advantages. Curved second-order grating SE-DFB lasers can reach a maximum continuous output power of 73 watts. The development of high-order gratings and new materials like perovskite and organic semiconductor materials is expected to further decrease manufacturing costs and threshold currents. Photonic Crystal Diffraction structures provide lasers with high output power, beam quality, and narrow divergence angles. For instance, the Noda team at Kyoto University developed a large-area dual-lattice photonic crystal laser that delivered a continuous output power of 29 W and a divergence angle of less than 0.4°, surpassing previous limitations of surface-emitting semiconductor lasers in high-power fields. Meanwhile, Mirror-type HCSELs have simpler working principles, but their manufacturing processes are more complicated, and they have larger divergence angles which have not been given as much attention. Nonetheless, improving the low cost and high beam quality of HCSELs remains a significant challenge for researchers. Especially for SE-DFB structures and photonic crystal diffraction structures, key factors to promote further development are the expansion of chip area, the improvement of surface coupling efficiency (such as buried second-order gratings), the suppression of high-order modes, and the increase of output power to broaden the laser's applications in fields such as material processing, laser medicine, nonlinear optics and optical communication. In the future, a high-power level cavity laser light source with high beam quality and high slope efficiency is worth anticipating and will have vast potential in various industries.”

Round 2
Reviewer 4 Report
Attached please find my comments.

Also shown in my attached comments.
Author Response
Thank you very much. Your comments are very valuable and helpful for improving our paper, as well as the important guiding significance to our future research. We have revised the manuscript according to your comments. Our point-by-point response is attached below.
Section1: something misleading.
- a) In line 105-106, “PCSELs can be classified as either horizontal cavity structures or vertical cavity structures based on cavity structure.” is misleading. I am afraid that Prof. Noda cannot agree on this point. PCSEL is coined as a special acronym and Ref. [22] has nothing to do with PCSEL.
Response a): Thank you for constructive and useful advice. We have fixed this error and we have changed the sentence to: “Photonic crystal structures have applications in both horizontal cavity lasers and vertical cavity lasers[22].” (Page 3)
- b) In line 132-133, “… first-order diffracted light wave will reflect back in the +x direction …” is somewhat ambiguous. Please check if it is correct. Moreover, Ref. [26] in line 136 is wrongly cited. It has nothing to do with PCSEL.
Response b): Thanks a lot for your guidance on this. We have changed the sentence to: “Furthermore, since it meets the first-order Bragg diffraction condition, light waves are diffracted into the ±x direction as well. Ultimately, the light waves propagating in these four directions interact, leading to the formation of a two-dimensional standing wave and a large-area photonic crystal cavity.” (Page 4) And we have replaced the reference with: “This phenomenon can be used to produce a photonic crystal surface emitting laser with a large area of light output and a small divergence angle [26] .” General recipe to realize photonic-crystal surface-emitting lasers with 100-W-to-1-kW single-mode operation | Nature Communications
- c) In line line361-363, “By employing simple nanoimprint … through system’s optical mode analysis.” is somewhat strange. Experiment and simulation are mixed in the same sentence.
Response c): Thanks a lot for your guidance on this. We have changed the sentence to: “Lastly, a simple nanoimprint was achieved on the perovskite film using the preparation schematic presented in Figure 10. Furthermore, cavity design is obtained by analyzing the optical mode of the system, and by adjusting the solution concentration, the effective refractive index of the waveguide mode can be controlled.” (Page 10)
- d) In line 443, “lack of 2D coupling” is described without explanation. Since both Meier and Imada used triangular lattice structures, the authors should elucidate the reasons behind.
Response d): Thanks a lot for your guidance on this. We have explained this: “Meier's group utilized the band edge of the triangular-lattice photonic crystal, but did not consider the Γ-point of the photonic crystal band edge. Consequently, there was a lack of two-dimensional coupling of light in the photonic crystal resonator, resulting in their device not exhibiting coherent two-dimensional (2D) oscillations.”.(Page 13)
- e) In line 448-450, “one-thousand times” and “two order of magnitude” were used but without metrics. Please consider rewriting it. Moreover, digital number associate with “M2” is missing.
Response e): Thanks a lot for your guidance on this. We have made the appropriate additions to the metrics. And we have added the value of “M2”. (Page 13)
- f) In line 521-522, what do you mean by “the contribution of the resonator”? In line 522-525, there are grammatical errors and bad punctuation; moreover, the sentence is too long and hard to read.
Response f): Thanks a lot for your guidance on this. We have changed the sentence to: “It is apparent that the resonant cavity's power consumption is relatively high when the slope efficiency is low.” And we have shortened the second sentence: “Thus, optimizing the radiation constant, tuning the number of quantum wells, and suppressing internal absorption loss can reduce the threshold current density of PCSELs. This will potentially lower heat generation within the device and further increases its maximum output power.” (Page 15)
Section2: imprecise description
- a) In line 243, “… at approximately 89 mm, …” is not complete. I suggest adding “in waveguide width” after 89 m
Response a): Thank you for reminding us. We have added “in wavelength” to the text. (Page 7)
- b) In line 443, Ref. [31] was contributed by not only Noda’s team but also Hamamatsu’s.
Response b): Thank you for reminding us. We have added “Hamamatsu’s team” to the text. (Page 13)
- c) In line 457, “… resulting in the elimination of …” is somewhat strange in context.
Response c): Thank you for reminding us. We have changed the sentence to: “Compared to a VCSEL, this laser achieves surface emission through photonic crystal diffraction, circumventing the growth cost of DBR. Additionally, the enhanced output power is a result of the photonic crystal's flat-band structure and an extra feedback mechanism.” (Page 13)
- d) In line 486, what do you mean by “controlled dimensions”.
Response d): Thank you for reminding us. We have removed this misleading sentence. (Page 14)
- e) In line 488-489, “… generating the same optical gain of the side-cavity mode …” is ambiguous. Please explain or rewrite it.
Response e): Thank you for reminding us. We have changed the sentence to: “By providing optical gain at the frequency of the band-edge cavity mode created in the photonic crystal, the device achieves in-plane wide-area coherent oscillation.” (Page 14)
- f) In line 492-394, “The proximity of high-order modes … than for the fundamental mode …” may need English language editing.
Response f): Thank you for reminding us. We have changed the sentence to: “The higher-order modes' proximity to the edges leads to a faster increase in edge losses compared to the fundamental modes, thereby resulting in larger threshold gain margins.” (Page 14)
Section3: redundant, typographic and erroneous descriptions
- In line 178, “pulse working” should be replaced by “pulsed operating”.
Response a): Thank you for your advice. We have changed “pulse working” to “pulsed operating”. (Page 6)
- In line 196-197, “They utilized ITO as the p-type cladding layer and the transparent ITO as the cladding layer …” seems redundant in context.
Response b): Thank you for reminding us. We have removed the redundant descriptions. (Page 6)
- c) In line 221, “secondary” should be replaced by “second-order”.
Response c): Thank you for reminding us. We have changed “secondary” to “second-order”. (Page 7)
- d) In line 249, “continuously in wave mode” should be replaced by “in continuous-wave mode”.
Response d): Thank you for reminding us. We have corrected “continuously in wave mode” to “in continuous-wave mode”. (Page 7)
- e) In line 294, “eveloped” is a typo.
Response e): Thank you for reminding us. We have corrected “eveloped” to “developed”. (Page 9)
- f) In line 460, “… a wavelength of 1.3 μm while under …” contains a typo.
Response f): Thank you for reminding us. We have changed the sentence to: “As a result of the study, a surface-emitting laser operating at a continuous wave with a wavelength of 1.3 μm at room temperature was developed. This device has an output of 13.3 mW of continuous power and 150 mW of pulsed power.” (Page 13)
- g) In line 463, “array of uses” should be replace with “uses of array”.
Response g): Thank you for reminding us. We have corrected “array of uses” to “uses of array”. (Page 13)
- h) In line 474, “dual-lattice structure” should be replaced by “dual-lattice PCSEL”.
Response h): Thank you for reminding us. We have corrected “dual-lattice structure” to “dual-lattice PCSEL”. (Page 14)
- i) In line 591, “CaAs” is a typo.
Response i): Thank you for reminding us. We have corrected “CaAs ” to “GaAs”. (Page 18)
- j) In line 605-606, “… they integrated … by integrating …” seems redundant.
Response j): Thank you for reminding us. We have removed the redundant descriptions. (Page 18)
- k) In line 633, “8.4 dB” should be replaced by “-8.4dB”. Please check it.
Response k): Thank you for reminding us. We have corrected “8.4 dB” to “-8.4dB”. (Page 19)
- l) In line 640, the acronym “SiP” is not defined in the text when they are used for the first time.
Response l): Thank you for reminding us. We have added the full name when it is used for the first time. (Page 19)

Round 3
Reviewer 4 Report
The revised manuscript is now much improved and can be accepted.
It could be much better if it can be polished by native speaker.